# Learning Fused State Representations for Control from Multi-View Observations

**Zeyu Wang** [* 1]  **Yao-Hui Li** [* 1]  **Xin Li** [† 1 2]  **Hongyu Zang** [3]  **Romain Laroche** [4]  **Riashat Islam** [5]

## Abstract

Multi-View Reinforcement Learning (MVRL) seeks to provide agents with multi-view observations, enabling them to perceive environment with greater effectiveness and precision. Recent advancements in MVRL focus on extracting latent representations from multiview observations and leveraging them in control tasks. However, it is not straightforward to learn compact and task-relevant representations, particularly in the presence of redundancy, distracting information, or missing views. In this paper, we propose **M**ulti-view **F**usion **S**tate for **C**ontrol (**MFSC**), firstly incorporating bisimulation metric learning into MVRL to learn task-relevant representations. Furthermore, we propose a multiview-based mask and latent reconstruction auxiliary task that exploits shared information across views and improves MFSC's robustness in missing views by introducing a mask token. Extensive experimental results demonstrate that our method outperforms existing approaches in MVRL tasks. Even in more realistic scenarios with interference or missing views, MFSC consistently maintains high performance. The project code is available at https://github.com/zpwdev/MFSC.

## 1. Introduction

In robotic manipulation tasks, acquiring accurate 3D scene information, including understanding target position, orientation, occlusions, and stacking relationships among objects in complex environments, is crucial for effective grasping and interaction with objects (Ze et al., 2023; Akinola et al., 2020). However, directly using 3D inputs is often bottle-necked by high data collection costs and increased computational complexity. MVRL has recently emerged as a promising alternative, leveraging 2D observations from multiple perspective cameras to enhance spatial perception while reducing computational demands (Li et al., 2019). Despite these advantages, MVRL still struggles with two common challenges: i) *the difficulty in learning and fusing compact representation from redundant and high-dimensional multi-view observations*, and ii) *the problem of dealing with missing or interfering views in real-world scenarios*.

Chen et al. (2021) averages keypoint estimations from different camera views to obtain a single world coordinate prediction. Hwang et al. (2023) learns distinct representations for each view using a VAE-based architecture and derives the final fused representation for MVRL via a weighted fusion mechanism. Jangir et al. (2022) introduces the use of a cross-view attention mechanism, offering a more flexible approach for representation fusion. However, these multi-view representation learning methods lack explicit mechanisms to focus on task-relevant features, often incorporating irrelevant information into the final representation. Recent works on bisimulation metric learning (Castro et al., 2021; Zhang et al., 2021; Zang et al., 2022) capture task-relevant representations by measuring behavioral similarities between states based on their rewards and transition distributions, and appear to be a promising solution to mitigate this issue.

In this work, we integrate the self-attention mechanism with bisimulation metric, offering a powerful approach to multi-view representation learning and fusion. Inspired by BERT (Devlin, 2018) and ViT (Dosovitskiy, 2020), a learnable fusion [state] embedding for MVRL tasks is introduced to avoid biases from fixed views, and dynamically capture the optimal fusion of multi-view information. By aligning the fusion of multi-view representations based on the similarity of task-relevant information, we enable robust multi-view integration and efficient control. However, bisimulation metric learning heavily depends on rewards to fuse state representations, which may cause it to overlook fine-grained information or interactions unique to each view not directly reflected in the reward signal. Secondly, bisimulation assumes complete and high-quality observations, making it vulnerable to scenarios where observations are incomplete, as it lacks mechanisms to compensate for missing views.

---

[*]Equal contribution, order is determined by a random drawing. [1]Beijing Institute of Technology, Beijing, China [2]Key Laboratory of Symbolic Computation and Knowledge Engineering of Ministry of Education, Jilin University, China [3]Meituan, Beijing, China [4]Wayve, London, UK [5]Mila - Quebec AI Institute, Montreal, Canada. [†]Correspondence to: Xin Li <xinli@bit.edu.cn>.

*Proceedings of the 42nd International Conference on Machine Learning*, Vancouver, Canada. PMLR 267, 2025. Copyright 2025 by the author(s).

Table 1: Comparison of different MVRL algorithms.

| Method | Masking | Applying Demos | Reconstruction | Framework | Multi-view Methods |
|---|---|---|---|---|---|
| Keypoint3D (Chen et al., 2021) | ✗ | ✗ | Pixel | Encoder-Decoder | 3D keypoints |
| LookCloser (Jangir et al., 2022) | ✗ | ✗ | ✗ | Encoder only | Cross-attention |
| F2C (Hwang et al., 2023) | ✗ | ✗ | ✗ | Encoder only | CVIBs |
| MV-MWM (Seo et al., 2023b) | Feature+View | ✓ | Pixel | Encoder-Decoder | Self-attention |
| MVD (Dunion & Albrecht, 2024) | ✗ | ✗ | ✗ | Encoder only | Contrastive learning |
| **MFSC** (ours) | Pixel | ✗ | Latent | Encoder only | Self-attention+Bisimulation |

To overcome these limitations, we propose an enhanced approach that integrates multiview-based masking and latent reconstruction along with bisimulation metric learning. The bisimulation-driven reconstruction helps ensure that the learned state representations are not only aligned with the task's global objectives but also retain crucial details that are specific to each individual view. Furthermore, the reconstruction process encourages representation learning to leverage cross-view dependencies to recover the masked information, rather than simply depending on the reward signal alone for state representation learning. Through the auxiliary task of multiview-based masking and latent reconstruction, we naturally introduce a learnable mask token to replace missing views, bolstering the model's capability to handle missing views. The key contributions of our work are summarized below:

- We propose a novel MVRL algorithm MFSC, which is the first to incorporate bisimulation into the MVRL framework. This integration enables the extraction of task-relevant representations from redundant multi-view observations. Our method can be combined with any existing RL algorithms with minimal adjustments.

- We propose a multiview-based masking and latent reconstruction auxiliary task that significantly enhances multi-view representation fusion. By introducing a learnable mask token, our method improves robustness against missing views.

- Extensive experiments on motion and robotic manipulation tasks demonstrate that our method outperforms existing MVRL algorithms, consistently achieving superior performance even in challenging scenarios with missing or noisy views.

## 2. Related Work

**Multi-View Representation Learning.** Multi-view representation learning uses multiple sources (or views) of a shared context. These views may include multiple camera views of the same scene (as in our work), as well as multi-modal inputs (e.g., audio and video) or synthetic views of the unimodal measurements (e.g., time-sequenced data). Li et al.

(2018) categorized existing methods into two approaches: representation alignment and representation fusion. Representation alignment aims to capture the relationships among multiple different views through feature alignment, such as minimizing the distance between representations of different views (Feng et al., 2014; Li et al., 2003), maximizing similarity between views (Bachman et al., 2019; Frome et al., 2013), and maximizing the correlation of variables across views (Andrew et al., 2013). Representation fusion integrates representations from different views to form a compact representation for downstream tasks (Geng et al., 2022; Xie et al., 2020; Karpathy & Fei-Fei, 2015). Both strategies aim to harness the complementary information provided by multiple views for more comprehensive data representation. In this work, we propose a multi-view representation fusion method designed for control. Our approach leverages the properties of sequential decision-making tasks and multi-view data to achieve task-relevant and robust representation fusion.

**Multi-View Reinforcement Learning.** Effective state representation learning in MVRL aims to transform high-dimensional observations into a compact latent space. Li et al. (2019) propose a VAE-based algorithm that minimizes Euclidean distance between states encoded from different views, assuming a consistent primary view. Keypoint3D (Chen et al., 2021) uses 3D reconstruction to learn keypoints from third-person views, requiring camera calibration. Lookcloser (Jangir et al., 2022) introduces a cross-view attention mechanism for aggregating egocentric and third-person representations without calibration, though at higher computational cost. F2C (Hwang et al., 2023) uses conditional variational information bottlenecks (CVIBs) to model state space, showing robustness to missing views. MV-MWM (Seo et al., 2023b), based on Dreamer-v2 (Hafner et al., 2021), applies dual masking and pixel-level reconstruction tasks during pretraining. MVD (Dunion & Albrecht, 2024) employs contrastive learning to disentangle multi-view representations for control tasks. MOSER (Wan et al., 2024) is a model-based approach that actively seeks the optimal perspective for learning task representations under multiple views to enhance performance. For a detailed comparison of MFSC with other closely related algorithms, refer to Table 1 and Appendix B.2.

# 3. Preliminaries

This section establishes the foundational concepts underlying our method, i.e., the Multi-View Markov Decision Process (MV-MDP) and the bisimulation metric.

## 3.1. Multi-View Markov Decision Processes

A Multi-View Markov Decision Process is defined as the following tuple $\mathcal{M} = \langle \mathcal{S}, \mathcal{A}, \vec{\mathcal{O}}, \mathcal{P}, \Omega, \mathcal{R}, \gamma \rangle$. $\mathcal{S}$ represents the set of states $s$ in the environment, $\mathcal{A}$ is a set of actions $a$, $\vec{\mathcal{O}} = \{\mathcal{O}^k\}_{k=1}^K$ represents the set of $K$ observations $\vec{o} = \{o^k\}_{k=1}^K$. With the assumption that each multi-view observation $\vec{o} \in \vec{\mathcal{O}}$ uniquely determines its generating state $s \in S$, we can obtain the latent state regarding its multi-view observation by a projection function $\phi(\vec{o}) : \vec{\mathcal{O}} \rightarrow \mathcal{S}$. Therefore, $s$ and $\phi(\vec{o})$ can be used interchangeably. $\mathcal{P}(s'|s, a) = \Pr(s_{t+1} = s'|s_t = s, a_t = a)$ is the transition dynamics distribution. The corresponding transition function under the multi-view observation space is defined $\vec{o}' \sim \hat{\mathcal{P}}(\vec{o}'|\vec{o}, a)$, where $\hat{\mathcal{P}}(\vec{o}'|\vec{o}, a) = \Omega(\vec{o}'|s') \mathcal{P}(s'|s, a)$ and $\Omega(\vec{o}|s) = \prod_{k=1}^K \Pr(o_t^k = o^k|s_t = s)$ is the probability distribution of joint observation. $\mathcal{R}(s, a) \in \mathbb{R}$ is the immediate reward function for taking action $a$ at state $s$, $\gamma \in [0, 1)$ is the discount factor. The goal of the agent is to find the optimal policy $\pi(a|s)$ to maximize the expected reward: $E_{s_0, a_0, \dots} [\sum_{t=0}^{\infty} \gamma^t r(s_t, a_t)]$.

## 3.2. Bisimulation

Bisimulation is used to formalize the concept of state similarity. The core principle is that two states are deemed equivalent if their future behaviors, i.e. state transitions and rewards given the same actions, are indistinguishable under the same policy. Conventional bisimulation metric needs to compute the Wasserstein distance over the transition distributions across all actions, which is computationally expensive. Instead, Castro (2020) developed $\pi$-bisimulation metric which removes the requirement of considering all actions and only needs to consider the actions induced by a policy $\pi$. Castro et al. (2021) introduced the MICo update operator as a mathematical tool for constructing a $\pi$-bisimulation metric. This metric evaluates state similarity by focusing exclusively on the actions dictated by a specific policy $\pi$, rather than considering the entire action space.

**Definition 3.1.** ($\pi$-bisimulation metric (Castro, 2020)). Let $\mathcal{M}$ be the set of all measures over $S$. The operator $\mathcal{F}^\pi : \mathcal{M} \rightarrow \mathcal{M}$ is defined as:

$$\mathcal{F}^\pi(g)(s_i, s_j) = (1-c) \cdot |r_{s_i}^\pi - r_{s_j}^\pi| + c \cdot \mathcal{W}(g)(\mathcal{P}_{s_i}^\pi, \mathcal{P}_{s_j}^\pi),$$

where $s_i, s_j \in S$, $r_{s_i}^\pi = \sum_{a \in A} \pi(a|s_i) r_{s_i}^a$, $\mathcal{P}_{s_i}^\pi = \sum_{a \in A} \pi(a|s_i) \mathcal{P}_{s_i}^a$, and $\mathcal{W}(g)$ denotes the Wasserstein distance between distributions, computed with cost function

$g$. $c$ controls the trade-off between prioritizing immediate reward differences and accounting for long-term future state dynamics.

**Theorem 3.2.** *Castro (2020) The operator $\mathcal{F}^\pi$ has a unique fixed point $g_\sim^\pi$ and $g_\sim^\pi$ is a $\pi$-bisimulation metric.*

# 4. Theoretical Analysis

MVRL introduces the challenge of dealing with observations that may contain redundant or irrelevant information across different views. To address this, we employ bisimulation metrics to capture task-relevant information from multi-view observations. This section formalizes our theoretical framework, extending the bisimulation metric from single-view RL to multi-view setting and demonstrating its effectiveness in simplifying MVRL tasks.

## 4.1. Bisimulation Metric in Multi-View Setting

Formally, we define the bisimulation metric for policy $\pi$ on a multi-view setting as:

$$\mathcal{F}^\pi G(\vec{o}_i, \vec{o}_j) = (1-c) \cdot |r_{\vec{o}_i}^\pi - r_{\vec{o}_j}^\pi| + c \cdot \mathbb{E}_{\vec{o}_i' \sim \hat{\mathcal{P}}_{\vec{o}_i}^\pi, \vec{o}_j' \sim \hat{\mathcal{P}}_{\vec{o}_j}^\pi} [G(\vec{o}_i', \vec{o}_j')], \quad (1)$$

where $\vec{o}_i, \vec{o}_j \in \mathcal{O}^1 \times \mathcal{O}^2 \times \cdots \times \mathcal{O}^k$, and the operator $\mathcal{F}^\pi$ has a unique fixed point $G^\pi$. Learning the exact bisimulation metric is computationally challenging. In this paper, we employ a "base metric" by setting $G^\pi$ as a simple cosine distance and guide the representation learning, a learnable aggregator $\phi$, to align with the behavioral pattern demonstrated in Eq.(1). Consequently, Eq.(1) is rewritten as:

$$\mathcal{F}^\pi G^\pi(\phi(\vec{o}_i), \phi(\vec{o}_j)) = (1-c) \cdot |r_{\phi(\vec{o}_i)}^\pi - r_{\phi(\vec{o}_j)}^\pi| + c \cdot \mathbb{E}_{\vec{o}_i' \sim \hat{\mathcal{P}}_{\vec{o}_i}^\pi, \vec{o}_j' \sim \hat{\mathcal{P}}_{\vec{o}_j}^\pi} [G^\pi(\phi(\vec{o}_i'), \phi(\vec{o}_j'))]. \quad (2)$$

The aggregator $\phi$ maps multi-view observations into a more compact space of summarizations $\mathcal{Z}$, defined as: $\phi : \mathcal{O}^1 \times \mathcal{O}^2 \times \cdots \times \mathcal{O}^k \rightarrow \mathcal{Z} \in \mathbb{R}^d$. By clustering observations to be similar under the bisimulation metric, the original MVRL problem can be approximated as solving a latent MDP $\bar{\mathcal{M}} = \langle \mathcal{Z}, \mathcal{A}, \bar{\mathcal{P}}, \Omega, \bar{\mathcal{R}}, \gamma, p_0 \rangle$.

## 4.2. Latent MDP Construction Assumptions

We establish the following assumption to support the construction of the latent MDP:

**Assumption 4.1.** There exists a latent space $\mathcal{Z}$ with dimension lower than that of $\mathcal{O}^1 \times \mathcal{O}^2 \times \cdots \times \mathcal{O}^k$, and a bounded approximation error $\eta > 0$, such that the induced latent MDP $\bar{\mathcal{M}} = \langle \mathcal{Z}, \mathcal{A}, \bar{\mathcal{P}}, \Omega, \bar{\mathcal{R}}, \gamma, p_0 \rangle$ satisfies:

1. **Markovian Property**: $\bar{\mathcal{P}}(z'|z, a) = \int_{\vec{o}'} \hat{\mathcal{P}}(\vec{o}'|\vec{o}, a)$ for all $\vec{o} = (o_1, o_2, ..., o_k) \in \mathcal{O}^1 \times \mathcal{O}^2 \times \cdots \times \mathcal{O}^k$ such that $\phi(\vec{o}) = z \in \mathcal{Z}$, and same for $\bar{\mathcal{R}}$.

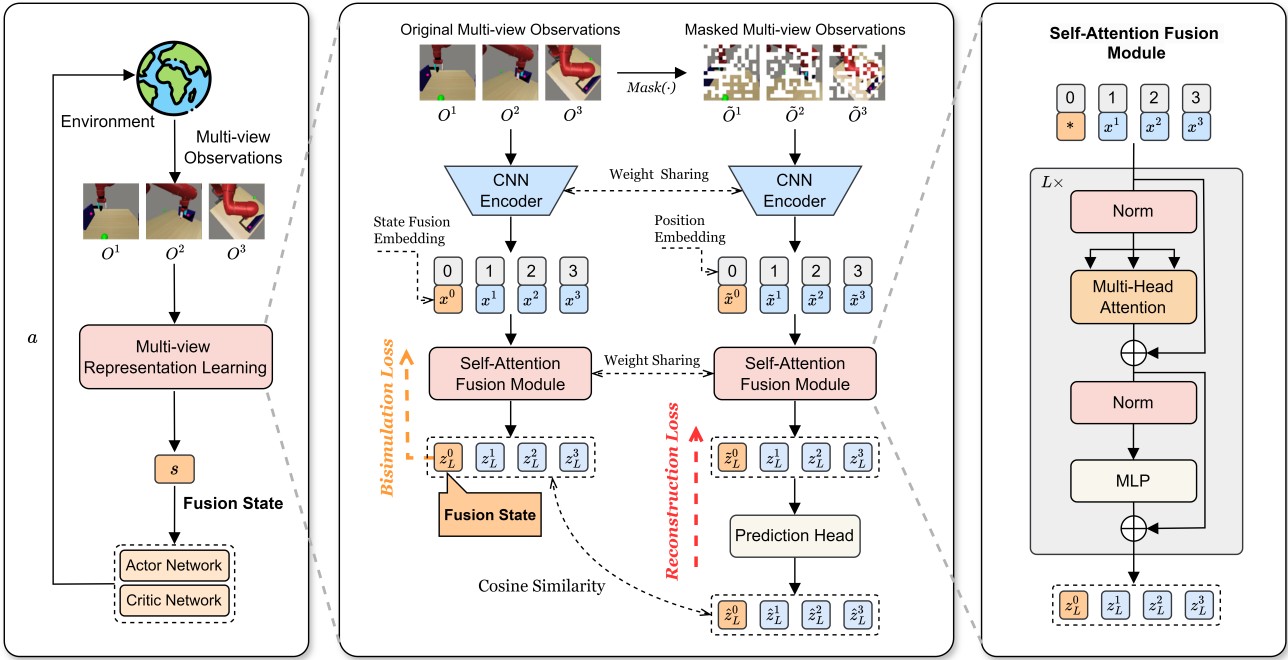

Figure 1: Framework of MFSC. (a) The **left** part illustrates the process of MVRL, where the agent receives observations from multiple views, learns a fused latent state, and interacts with the environment through an actor-critic framework. (b) The **middle** part provides a detailed overview of the MFSC architecture. Each view is encoded into a latent embedding via a CNN, followed by state fusion using the Self-Attention Fusion Module. Metric learning is guided by bisimulation loss, and a mask-based self-supervised auxiliary task is employed to enhance the model's cross-view learning capabilities. (c) The **right** part presents the inner workings of the Self-Attention Fusion Module, which integrates embeddings from different views through attention mechanisms to produce a unified state representation.

2. **Value Function Approximation**: For every $\vec{o} \in \vec{\mathcal{O}}$, there exists $z \in \mathcal{Z}$ such that $|V^*(\vec{o}) - V^*(z)| \leq \eta$.

This assumption ensures that $\mathcal{Z}$ retains sufficient information for value estimation while reducing redundancy.

### 4.3. Bounding Value Function Differences

To quantify the impact of the above transformation, we derive the following lemma:

**Lemma 4.2** (Value Function Bound). *For a latent MDP $\bar{\mathcal{M}}$ constructed by an aggregator $\phi : \mathcal{O}^1 \times \mathcal{O}^2 \times \cdots \times \mathcal{O}^k \to \mathcal{Z}$ that clusters multi-view observations in a $\varepsilon$-neighborhood. The difference between the optimal value functions of the original and latent MDPs is bounded as:*

$$|V^*(\vec{o}) - V^*(\phi(\vec{o}))| \leq \frac{2\varepsilon}{(1-\gamma)(1-c)}, \qquad (3)$$

*where $c \in [0, 1)$ is the trade-off parameter in the bisimulation metric, and $c \geq \gamma$.*

The proof can be found in the Appendix A. Lemma 4.2 serves to establish a bound on the difference between the optimal value functions of multi-view observations and their

corresponding clusters in a simplified MDP. By leveraging the learned aggregator, we can effectively reduce the complexity of the multi-view MDP's state space while maintaining a predictable level of accuracy in value function estimation.

## 5. Learning Fused State Representations

In this section, we detail how the aggregator $\phi$ is constructed to produce fused state representations. Fig.1 outlines the overall framework, which comprises two key components: **(i) Self-Attention Fusion Module**, which integrates bisimulation metrics to guide the aggregator in extracting task-relevant features from multi-view observations; and **(ii) Multiview-based Masking and Latent Resconstruction**, which serves as an auxiliary task to further promote cross-view state aggregation.

### 5.1. Self-Attention Fusion Module

In this section, we will introduce how bisimulation can be incorporated into multi-view representation learning. Specifically, our method comprises two core submodules: **(i) Convolutional Feature Embedding**, generating embeddings

of the original high-dimensional multi-view observations; **(ii) Self-Attention Fusion Module**, integrating multi-view representations based on bisimulation metric.

**Convolutional Feature Embedding.** The feature encoding module uses a Convolutional Neural Network (CNN) encoder to encode single-view image observations into fixed-dimensional embeddings. Given a multi-view observations $\vec{o} = \{o^1, o^2, \ldots, o^k\}$, where $o^i \in \mathbb{R}^{H \times W \times C}$, the CNN encodes each image into a single-view representation $x^i$, where $x^i \in \mathbb{R}^d$.

**Self-Attention Fusion Module.** Similar to the [class] token used in BERT (Devlin, 2018) and ViT (Dosovitskiy, 2020), we prepend a learnable fusion [state] embedding $x^0 \in \mathbb{R}^d$ to the sequence of multi-view embedded representations. The state fusion representation $x^0$, trained via self-attention and the bisimulation metric, serves as the final fused representation of multi-view observations and is used for downstream RL tasks. Additionally, akin to ViT, a trainable positional embedding 1D tensor $E_{pos}$ is incorporated into the multi-view embeddings to encode view-specific information:

$$z_0 = [x^0, x^1, x^2, \ldots, x^k] + E_{pos}, \ E_{pos} \in \mathbb{R}^{(k+1) \times d}. \quad (4)$$

The embedded sequence is then fed into the Self-Attention Fusion Module. Specifically, the Self-Attentioni Fusion Module consists of $L$ attention layers. Each layer is composed of a Multi-Headed Self-Attention (MHSA) layer, a layer normalization (LN), and Multi-Layer Perceptron (MLP) blocks. The process can be described as follows:

$$z'_\ell = \text{MHSA}(\text{LN}(z_{\ell-1})) + z_{\ell-1}, \quad \ell = 1 \ldots L \quad (5)$$

$$z_\ell = \text{MLP}(\text{LN}(z'_\ell)) + z'_\ell. \quad \ell = 1 \ldots L \quad (6)$$

The output after $L$ attention layers is $z_L = \{z_L^0, z_L^1, \ldots, z_L^k\}$, where $z_L^0$ represents the final state fusion embedding. To extract task-relevant representations from multi-view observations, bisimulation metric learning is integrated into the state fusion process. Specifically, the bisimulation metric for a given policy $\pi$ (Eq.2) employs the measurement $G$, as defined in SimSR (Zang et al., 2022). Here, $G$ is based on cosine distance, which offers lower computational complexity compared to the Wasserstein distance while effectively mitigating representation collapse.

In RL, the critic in actor-critic algorithms like SAC (Haarnoja et al., 2018), can be decomposed into two function approximators, $\psi$ and $\phi$, with parameters $\theta$ and $\omega$ respectively: $Q_{\theta,\omega} = \psi_\theta (\phi_\omega (\vec{o}))$. Here, $\psi_\theta$ serves as the value function approximator, while $\phi_\omega$ is the state aggregator, with the goal of aligning the distances between representations to match the cosine distance. Therefore, the parameterized representation distance $G_{\phi_\omega}$ can be defined as an approxi-

mation to the original observation distance $G^\pi$:

$$G^\pi(\vec{o}_i, \vec{o}_j) \approx G_{\phi_\omega}(\vec{o}_i, \vec{o}_j) := 1 - \cos_{\phi_\omega}(\vec{o}_i, \vec{o}_j)$$

$$= 1 - \frac{\phi_\omega(\vec{o}_i)^T \cdot \phi_\omega(\vec{o}_j)}{\|\phi_\omega(\vec{o}_i)\| \cdot \|\phi_\omega(\vec{o}_j)\|}. \quad (7)$$

Therefore, the objective of state fusion with bisimulation metric is:

$$\mathcal{L}_{fus} = \mathbb{E}_{(\vec{o}_i, r, a, \vec{o}'_i), (\vec{o}_j, r, a, \vec{o}'_j) \sim \mathcal{D}} \left( G_{\phi_\omega}(\vec{o}_i, \vec{o}_j) - G_\odot \right)^2.$$

$$\begin{cases} G_\odot = \left| r^\pi_{\vec{o}_i} - r^\pi_{\vec{o}_j} \right| + \gamma G_{\phi_\omega}(s'_i, s'_j) \\ s'_i \sim \mathcal{P} \left( \cdot \left| \phi_\omega(\vec{o}'_i), a \right. \right), s'_j \sim \mathcal{P} \left( \cdot \left| \phi_\omega(\vec{o}'_j), a \right. \right) \end{cases} \quad (8)$$

$\mathcal{P}$ is latent state dynamics model and $\mathcal{D}$ is the replay buffer. For detailed explanations of the latent state dynamics model and the reward scaling mechanism, please refer to the Appendices C.2 and C.3. By incorporating bisimulation metrics during state aggregation, our method is able to focus on the causal features that directly influence rewards, effectively integrating information from multi-view observations.

### 5.2. Multiview-based Masking and Latent Reconstruction

Although bisimulation learning provides an effective approach to aggregating task-relevant information across multiple views, it overlooks the local details within each view, as it focuses on global state transitions and task reward rather than the details of each individual view. To address these challenges, we introduce a multiview-based masking and latent reconstruction auxiliary task, fostering the learning of cross-view dependencies. The masking and reconstruction mechanism strengthens bisimulation's capacity to capture richer multi-view information, enabling more robust fusion and control.

Specifically, we randomly mask a portion of the original multi-view image observations $\{o^1, o^2, \ldots, o^k\}$. The masked observation sequences $\{\tilde{o}^1, \tilde{o}^2, \ldots, \tilde{o}^k\}$ are then processed through the CNN Encoder and the Self-Attention Fusion Module, resulting in a set of masked state embeddings $\{\tilde{z}_L^0, \tilde{z}_L^1, \ldots, \tilde{z}_L^k\}$. Motivated by the success of SimSiam (Chen & He, 2021) in self-supervised learning, we use an asymmetric architecture to calculate the distance between the reconstructed latent states and the target states. The state embeddings of the masked observations are passed through a prediction head to get the final reconstructed/predicted states $\{\hat{z}_L^0, \hat{z}_L^1, \ldots, \hat{z}_L^k\}$. We construct the reconstruction loss using cosine similarity, ensuring that the final predicted result closely approximates its corresponding target. The final objective function of masking and latent reconstruction can be formulated as:

$$\mathcal{L}_{rec} = 1 - \frac{1}{k+1} \sum_{i=0}^{k} \frac{(\hat{z}_L^i)^T \cdot z_L^i}{\|\hat{z}_L^i\| \cdot \|z_L^i\|}. \quad (9)$$

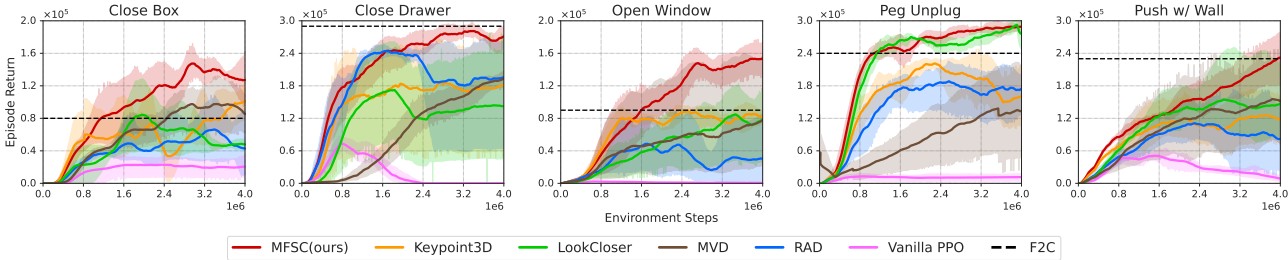

Figure 2: Performance comparison on five robotic arm manipulation tasks from Meta-World. All curves show the mean and its 95% Confidence Intervals (CIs) of performance across four independent seeds. The black dashed line represents the final convergence result of F2C in Meta-World.

The masking and latent reconstruction serves as an auxiliary task and is optimized together with the multi-view state fusion module. Thus, the overall loss function of MFSC is:

$$\mathcal{L}_{mfsc} = \mathcal{L}_{fus} + \lambda \mathcal{L}_{rec} \tag{10}$$

$\lambda$ represents the weighting coefficient for the two loss functions.

# 6. Experiment

Through our experiments, we aim to investigate the following questions: (1) How does the performance of MFSC compare to existing MVRL algorithms? (2) Has MFSC successfully captured task-relevant representations? And, how does the Self-Attention Fusion Module operate across different views? (3) To what extent can MFSC handle tasks with missing views? (4) How does the MFSC perform in the presence of noise interference? (5) What is the relative importance and contribution of each part of MFSC?

## 6.1. Setup

**Experimental Setup.** We evaluate our method on a set of 3D manipulation environments Meta-World (Yu et al., 2020) and a high degree of freedom 3D locomotion environment PyBullet's Ant (Coumans & Bai, 2022). These environments are originally designed for state-based RL, posing significant challenges for pixel-based RL. Furthermore, we also evaluate MFSC on the widely used single-view continuous control benchmark DMControl (Tassa et al., 2018), comparing it with recent visual RL baselines, as well as in the more realistic multi-view highway driving scenario, CARLA (Dosovitskiy et al., 2017). More details about the experiments can be found in the Appendix E.

For fair comparison, we adopt the same Meta-World experimental setup as described in Chen et al. (2021). Details regarding the specific task settings of Meta-World can be found in the Appendix D. PyBullet's Ant is highly dynamic, with different colors assigned to adjacent limbs to enhance

the agent's fine-grained understanding of movable joints and components. To further validate whether our algorithm can still capture task-relevant information, we also evaluate it in a more practical *'Ant No Color'* task, where no explicit visual cues are provided.

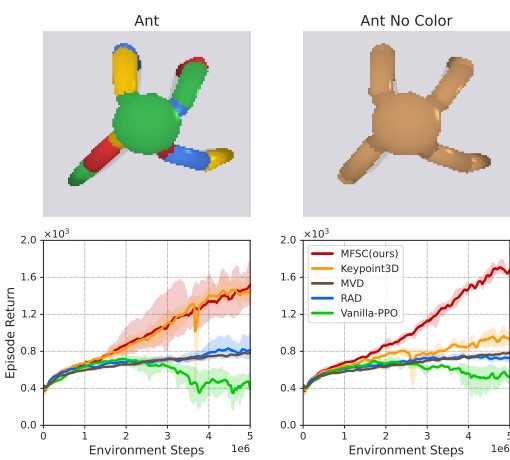

Figure 3: Performance on PyBullet's Ant.

**Baselines.** We compare MFSC with several baseline methods. All baselines, including MFSC, are implemented with PPO (Schulman et al., 2017). The baselines include: (1) Keypoint3D (Chen et al., 2021), which learns 3D keypoints from multiple third-person view cameras; (2) LookCloser (Jangir et al., 2022), which applies cross-attention between pairs of views to integrate multi-view information; (3) Fuse2Control (F2C) (Hwang et al., 2023), which employs an information-theoretic approach to learn a state space model and extract information independently from each view; (4) MVD (Dunion & Albrecht, 2024), a contrastive learning-based algorithm for disentangling multi-view representations. Additionally, for common RL algorithms, we stack images from all three views to form the observation: (5) RAD (Laskin et al., 2020a), which achieves high sample efficiency through data augmentation; (6) Vanilla PPO, the original PPO algorithm (Schulman et al., 2017).

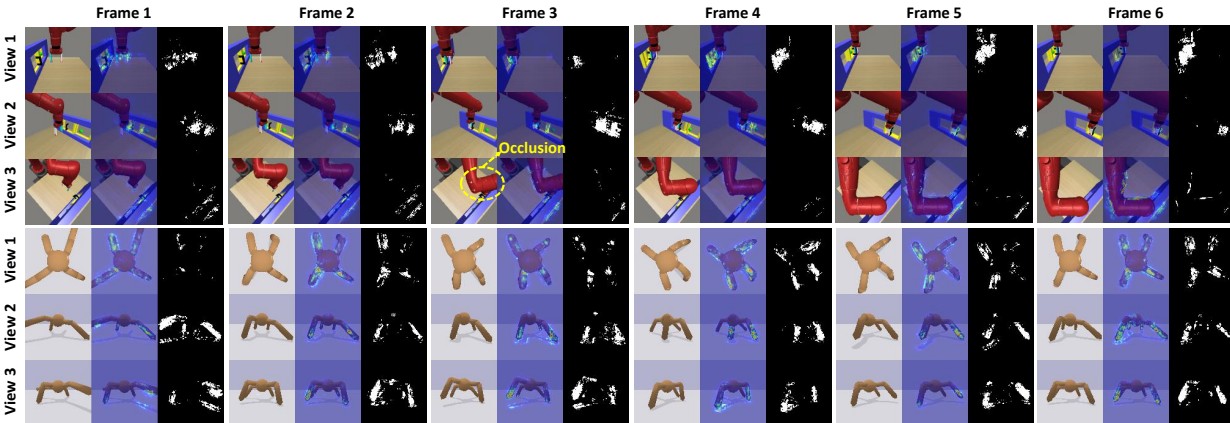

Figure 4: Visualization of multi-view fusion for task-relevant representations. The first column of each frame represents snapshots from different views. The second column of each frame displays Grad-CAM heatmaps for each view, generated from the task loss and normalized via min-max scaling. In the third column of each frame, we unify the gradient maps from all three views. We select the top 2.5% of gradient values across all views, highlight those corresponding pixels in white within each view, and show all other pixels in black.

## 6.2. Evaluation on Meta-World and PyBullet's Ant

**Meta-World.** As shown in Fig.2, our method consistently outperforms other MVRL algorithms across all five tasks, exhibiting significantly higher sample efficiency and more stable performance. In particular, Vanilla PPO shows scarcely any indication of performance improvement, indicating the difficulty in extracting meaningful fused representations without any special designs. While RAD integrates data augmentation and performs well in simpler tasks, it struggles to learn effective fused representations in more complex tasks, such as *'Open Window'* and *'Close Box'*. Keypoint3D demonstrates competitive performance in certain tasks, particularly in *'Close Box'*, but its overall training efficiency and final performance remain suboptimal. Look-Closer, which is also based on Transformer architecture, is also effective. However, its overall performance has not yet exceeded that of MFSC. Moreover, our method outperforms the previous SOTA algorithm, F2C, in terms of convergence performance across the majority of tasks.

**PyBullet's Ant.** As shown in Fig.3, the performance of our method in the *'Ant'* environment is comparable to Keypoint3D. During training, our algorithm exhibits stable and consistent performance improvement. Even in the more practical *'Ant No Color'* task, our method exhibits the highest performance, demonstrating its ability to effectively capture and fuse task-relevant information from multi-view observations. In contrast, methods based on contrastive learning and pixel-level reconstruction tend to overemphasize local variations, making it difficult to capture task-relevant information and leading to significant performance degradation when key color cues are removed.

## 6.3. Visualization and Analysis

To verify that MFSC is indeed capable of capturing task-relevant features, we utilize the Gradient-weighted Class Activation Mapping (Grad-CAM) technique (Selvaraju et al., 2017) to visualize the learned representations of MFSC. As shown in Fig.4 (middle column of each frame), MFSC consistently focuses on task-relevant features, such as the target position, robotic arm, or ant's legs, while minimizing focus on less relevant elements, like window edges or the ant's body. This analysis demonstrates MFSC's ability to effectively identify and extract task-relevant representations from each view.

In the *'Open Window'* task (first row), during the initial stage where the goal is to position the robotic arm correctly, all three views provide critical information, and the model distributes its attention accordingly. However, when the goal shifts to opening the window, occlusion in **View 3** leads the model to focus more on **View 1** and **View 2**, which contain more relevant details. In the *'Ant No Color'* task, the importance of information from all three views is relatively balanced, as shown by the uniform distribution of top-gradient pixels, indicating that MFSC allocates attention more evenly.

The visualization above demonstrates that MFSC can effectively extract and fuse task-relevant representations from multiple views. The fused representations enhance the performance of downstream RL tasks by offering a more robust and comprehensive understanding of the environment. By integrating information from diverse observational sources, MFSC enables more efficient policy learning and decision-making in complex control scenarios.

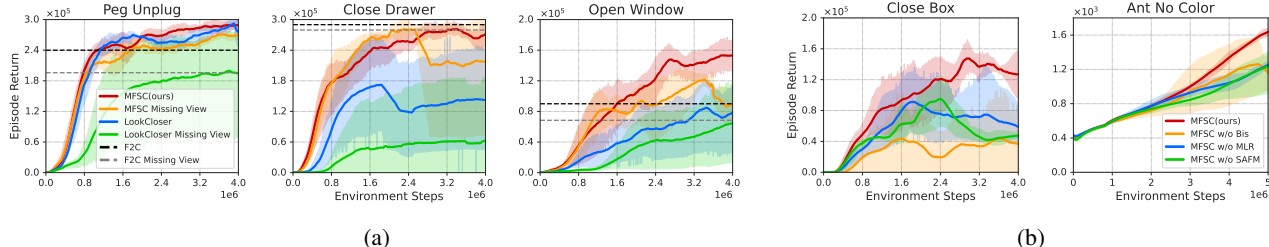

(a)                                 (b)

Figure 5: (a) Performance comparison of MFSC under full-view and missing-view. The black and gray lines represent the final convergence results of F2C. (b) Performance of ablation study.

## 6.4. Robustness Against Missing Views

To explore whether MFSC has the ability to counteract missing views, we systematically evaluate the performance of MFSC under conditions of missing views across three tasks from Meta-World. During training, we explicitly introduced a mask token and leveraged the masking and latent reconstruction to learn shared representations across views. In the testing phase, we simulate real-world scenarios with incomplete view information by randomly omitting observations from one view and replacing the missing view with the mask token. Fig.5(a) summarizes the performance comparison between MFSC and two other competitive MVRL algorithms, LookCloser and F2C, under both missing view and full view conditions. The results indicate that, although missing views can affect certain task-specific details, MFSC still exhibits decent performance in some tasks. Compared to LookCloser, the performance degradation of MFSC is relatively limited, and it even achieves performance comparable to the previous SOTA algorithm for missing views, F2C. This robustness to missing views is largely attributed to the introduction of multiview-based masking and latent reconstruction, which leverages cross-view dependencies to promote robust shared representation learning.

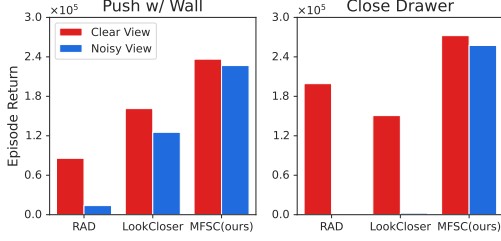

Figure 6: Performance on noisy view.

## 6.5. Robustness Against Noisy Views

To evaluate the robustness of MFSC under interference views, we introduce an additional disruptive view, composed of images entirely irrelevant to the task. As depicted in Fig.6, both RAD and LookCloser suffer notable performance decline when subjected to noise, particularly in the '*Closer Drawer*' task. Both simple concatenation of observations from different views (RAD) and fused representations derived through attention mechanisms (LookCloser) prove susceptible to the disruptive effects of noise. In contrast, MFSC preserves its high performance despite the introduction of irrelevant information.

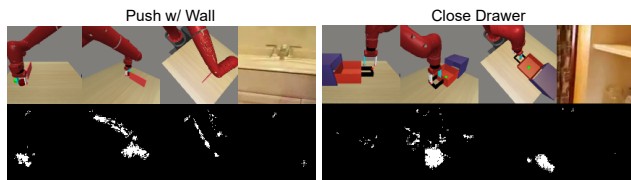

Figure 7: Partial visualization under the noisy view.

From the visualization analysis in Fig.7, by incorporating bisimulation into the representation learning process, MFSC continues to capture task-relevant features, thereby significantly enhancing its robustness to interference views. More details on visualization and ablation study are in the Appendix E.3

## 6.6. Ablation Study

Fig.5(b) illustrates the returns of the algorithm across two benchmarks, Meta-World and PyBullet's Ant, comparing three variations: MFSC, MFSC without bisimulation constraints ('MFSC w/o Bis'), MFSC without Masking and Latent Reconstruction ('MFSC w/o MLR'), and MFSC without Self-Attention Fusion Module ('MFSC w/o SAFM'), which means an average of the embeddings for each view.

MFSC (red line), as the complete method, achieves the highest returns throughout the process. In contrast, removing the bisimulation constraint in MFSC (orange line) not only significantly degrades performance but also increases variance, which highlights the importance of bisimulation in MFSC. 'MFSC w/o MLR' (blue line) performs better than 'MFSC w/o Bis' (orange line) but still falls short of the full MFSC method, emphasizing the significance of learning cross-view information. Finally, replacing the Self-Attention Fusion Module (green line) with feature averaging leads to a noticeable performance drop, demonstrating that the self-attention mechanism's dynamic integration of cross-view information is more effective than simple averaging.

# 7. Discussion

**Limitations and Future Work.** One limitation of our approach lies in its reliance on masking techniques to handle missing views. For certain views containing crucial information, accurately reconstructing the true environmental state remains challenging. This is due to the incomplete complementarity of information between multiple views, particularly in scenarios involving complex state transitions. To address this limitation, future research could explore the integration of state-space models (Lee et al., 2020) to better capture temporal dependencies, enabling more accurate state estimation even in the absence of certain views.

**Conclusion.** We propose a novel framework, Multi-view Fusion State for Control (MFSC), to address the challenge of representation learning in MVRL. MFSC integrates the self-attention mechanism with bisimulation metric learning, guiding the fusion of task-relevant representations from multi-view observations. Additionally, we introduce an auxiliary task, multiview-based masking and latent reconstruction, to capture cross-view information and enhance robustness to missing views. Experimental results demonstrate that MFSC effectively aggregates task-relevant details and shows robustness in scenarios with missing views. Finally, visualization analyses confirm the capability of MFSC to capture task-relevant information and dynamically fuse multi-view observations.

# Acknowledgements

The authors would like to thank anonymous reviewers for their constructive feedback, which has significantly contributed to enhancing the quality and clarity of this paper. This work was partially supported by the NSFC under Grants 92270125 and 62276024.

# Impact Statement

The proposed MFSC framework advances MVRL by tackling key challenges in learning task-relevant representations from multi-view observations. By combining bisimulation-based representation learning with a masking-based latent reconstruction auxiliary task, MFSC improves robustness and adaptability in environments characterized by redundant, distracting, or incomplete inputs. These improvements are particularly beneficial for applications such as autonomous robotics, self-driving systems and other scenarios where reliable perception is essential.

Ethical considerations include potential biases from sensor placement, modality imbalance, or observation dropout, which may affect policy fairness and generalizability. Future work should explore bias mitigation strategies and improve interpretability, particularly in safety-critical deployments.

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

# A. Proofs

**Lemma 4.2.** *Given a latent MDP $\bar{\mathcal{M}}$ constructed by a learned aggregator $\phi : \mathcal{O}^1 \times \mathcal{O}^2 \times \cdots \times \mathcal{O}^k \to \mathcal{Z}$ that clusters multi-view observations in a $\varepsilon$-neighborhood. The optimal value functions of the original MDP and the latent MDP are bounded as:*

$$|V^*(\vec{o}) - V^*(\phi(\vec{o}))| \leq \frac{2\varepsilon}{(1-\gamma)(1-c)}. \tag{11}$$

*Proof.* The proof follows straightforwardly from DBC (Zhang et al., 2021). From Theorem 5.1 in Ferns et al. (2004) we have:

$$(1-c)|V^*(\vec{o}) - V^*(\phi(\vec{o}))| \leq g(\vec{o}, \tilde{d}) + \frac{\gamma}{1-\gamma} \max_{u \in \mathcal{O}^1 \times \mathcal{O}^2 \times \cdots \times \mathcal{O}^k} g(u, \tilde{d}), \tag{12}$$

where $g$ is the average distance between a multi-view observation and all other multi-view observations in its equivalence class under the bisimulation metric $\tilde{d}$. By specifying a $\varepsilon$-neighborhood for each cluster of multi-view observations, we can replace $g$:

$$(1-c)|V^*(\vec{o}) - V^*(\phi(\vec{o}))| \leq 2\varepsilon + \frac{\gamma}{1-\gamma}2\varepsilon$$

$$|V^*(\vec{o}) - V^*(\phi(\vec{o}))| \leq \frac{1}{1-c}\left(2\varepsilon + \frac{\gamma}{1-\gamma}2\varepsilon\right)$$

$$= \frac{2\varepsilon}{(1-\gamma)(1-c)}.$$

As $\varepsilon \to 0$, the optimal value function of the aggregated MDP converges to the original value function. By defining a learning error for $\phi$, $\mathcal{L} := \sup_{\vec{o}_i, \vec{o}_j \in \mathcal{O}^1 \times \mathcal{O}^2 \times \cdots \times \mathcal{O}^k} \left| \||\phi(\vec{o}_i) - \phi(\vec{o}_j)\||_1 - \tilde{d}(\vec{o}_i, \vec{o}_j) \right|$, we can also update the bound in Lemma 4.2 to incorporate $\mathcal{L}$ : $|V^*(\vec{o}) - V^*(\phi(\vec{o}))| \leq \frac{2\varepsilon + 2\mathcal{L}}{(1-\gamma)(1-c)}$.

# B. Additional Related Work Discussion

In this section, we provide a more detailed discussion on state representation learning in RL and a detailed comparison between our method and other related work.

## B.1. State Representation Learning in RL

Well-constructed state representations enable agents to better understand and adapt to complex environments, thus improving task performance and decision-making efficiency. For example, methods such as CURL (Laskin et al., 2020b) and DrQ series (Kostrikov et al., 2020; Yarats et al., 2021a), leverage data augmentation techniques such as cropping and color jittering to enhance model generalization. However, their performance is highly dependent on the specific augmentations applied, leading to variability in results. Masking-based approaches (Seo et al., 2023b; Yu et al., 2022b; Seo et al., 2023a; Liu et al., 2022), selectively obscure parts of the input to mitigate redundant information and improve training efficiency. Although these methods show promise in filtering out irrelevant data, they carry the risk of unintentionally discarding task-critical information, potentially affecting overall agent performance. Bisimulation-based methods (Zhang et al., 2021; Zang et al., 2022) focus on constructing reward-sensitive state representations to ensure that states with similar values are close in the representation space, promoting sample efficiency and consistent decision-making. Another line of research explores the causal relationships between state representations and control (Wang et al., 2022; Lamb et al., 2022; Efroni et al., 2021; 2022; Fu et al., 2021). By analyzing the causal links between states and actions, these methods aim to improve agents' understanding and control of the environment, further optimizing RL performance.

## B.2. Comparison with Additional Related Work

**MV-MWM (Seo et al., 2023b).** MV-MWM proposed an RL framework that trained a multi-view masked autoencoder for representation learning and a world model to enhance the model-based RL algorithm's robustness to view variations in robotic manipulation tasks. MV-MWM adopts a dual masking strategy at both the convolutional feature and view levels, requiring an additional decoder to perform pixel-level reconstruction, which aims to recover task-specific details from raw observations. Our method applies masking at the pixel level but conducts reconstruction in the latent space, thereby avoiding

the reconstruction of task-irrelevant details and focusing on high-level task-related information. Furthermore, MV-MWM incorporates other robust components to enhance performance. For example, it is built on the powerful model-based RL baseline, Dreamer-v2 (Hafner et al., 2021), and utilizes expert demonstrations to guide policy learning. In contrast, our method does not employ any of these components. As MV-MWM requires additional expert demonstration data for training, we did not conduct a direct comparison with it in our work.

**MLR (Yu et al., 2022a).** Although both MLR and our method leverage a mask and latent reconstruction auxiliary task to learn more compact representations, there are notable differences in their implementation details. Firstly, MLR predicts complete state representations in the latent space from observations with spatially and temporally masked pixels. It prioritizes extracting inherent information from images, often overlooking task-relevant information. In contrast, our method integrates bisimulation metric learning, enabling the extraction of task-relevant information from multi-view observations. Secondly, our method employs the self-attention module as a fusion mechanism to learn a fused state representation that directly contributes to downstream RL tasks. Instead, MLR uses the self-attention mechanism merely as an auxiliary loss module to encourage the CNN encoder to capture temporal dependencies within sequences.

## C. Experimental Details

### C.1. Hyperparameters

Table 2 and 3 provide detailed information regarding the experimental setup and hyperparameter configurations. In the Meta-World and PyBullet environments, we utilize a latent representation size of 128, following the Keypoint3D framework outlined by Chen et al. (2021). In the CARLA and DMControl environment, we utilize a representation size of 50, adhering to the SAC framework described in Zhang et al. (2021). All networks in both the policy and the representation models are optimized using the Adam optimizer (Kingma, 2014), ensuring consistent performance across various environments.

Table 2: MFSC's hyperparameters, based on PPO.

| Hyperparameter | Meta-World | PyBullet |
|---|---|---|
| **General** | | |
| Batch size | 6400 | 16000 |
| Rollout buffer size | 100000 | 100000 |
| Frame stack | 1 | 2 |
| Num of views | 3 | 3 |
| Epochs per update | 8 | 10 |
| Discount | 0.99 | 0.99 |
| GAE lambda | 0.95 | 0.95 |
| Clip range | 0.2 | 0.2 |
| Entropy coefficient | 0.0 | 0.0 |
| Value function coefficient | 0.5 | 0.5 |
| Gradient clip | 0.5 | 0.5 |
| Target KL | 0.12 | 0.12 |
| Learning rate | $2 \times 10^{-4}$ | $2 \times 10^{-4}$ |
| State representation dimension | 128 | 128 |
| Weight of loss $\lambda$ | 1.0 | 1.0 |
| **MFSC** | | |
| Representation learning rate | $2 \times 10^{-4}$ | $2 \times 10^{-4}$ |
| Weight of fusion loss | 1.0 | 1.0 |
| Weight of reconstruction loss | 1.0 | 1.0 |
| Number of dynamics models | 5 | 5 |
| Mask ratio | 0.8 | 0.8 |
| Cube spatial size | $12 \times 12$ | $12 \times 12$ |
| Cube depth | 3 | 3 |
| Self-attention fusion module depth | 2 | 2 |

Table 3: MFSC's hyperparameters, based on SAC.

| Hyperparameter | Carla | DMControl |
|---|---|---|
| **General** | | |
| Batch size | 128 | 128 |
| Replay buffer size | 100000 | 100000 |
| Frame stack | 3 | 3 |
| Num of views | 3 | 4 |
| Discount | 0.99 | 0.99 |
| Critic Q-function soft-update rate | 0.01 | 0.01 |
| Critic target update frequency | 2 | 2 |
| Actor update frequency | 2 | 2 |
| Init temperature | 0.1 | 0.1 |
| Learning rate | $1 \times 10^{-3}$ | $2 \times 10^{-4}$ |
| Temperature learning rate | $1 \times 10^{-4}$ | $1 \times 10^{-4}$ |
| State representation dimension | 50 | 50 |
| Weight of loss $\lambda$ | 1.0 | 1.0 |
| **MFSC** | | |
| Representation learning rate | $2 \times 10^{-4}$ | $2 \times 10^{-4}$ |
| Weight of fusion loss | 1.0 | 1.0 |
| Weight of reconstruction loss | 1.0 | 0.5 |
| Number of dynamics models | 5 | 5 |
| Mask ratio | 0.5 | 0.5 |
| Cube spatial size | $12 \times 12$ | $12 \times 12$ |
| Cube depth | 3 | 3 |
| Self-attention fusion module depth | 2 | 2 |

## C.2. Latent State Dynamics Modeling

Following our approach, we develop an ensemble version of deterministic dynamics models $\mathcal{P}_k(\cdot|\phi_\omega(x), a)\}_{k=1}^{K}$. Unlike probabilistic dynamics models, our transition models are deterministic and their outputs are consistent with the encoder's output, both of which are subjected to $l_2$-normalization. Instead of using a probabilistic transition, we calculate the distance using cosine similarity. Specifically, at the training step, we update the parameters of the dynamics models based on the cosine similarity loss function:

$$\mathcal{L}_{dyn} = \frac{1}{K}\sum_{k=1}^{K}\left[1 - \frac{\mathcal{P}_k(\cdot|\phi_\omega(\vec{o}), a) \cdot \phi_\omega(\vec{o'})}{\|\mathcal{P}_k(\cdot|\phi_\omega(\vec{o}), a)\|\|\phi_\omega(\vec{o'})\|}\right], \tag{13}$$

where $i \in \{1, 2, \ldots, K\}$. Since deterministic models share the same gradient but are initialized randomly, they may still acquire different parameters after training. This ensemble model allows us to estimate the latent dynamics of the environment effectively while ensuring that the output remains consistent across the encoder and dynamics model. At the inference step, we randomly sample one of $K$ deterministic dynamics models to compute the transition to the next latent state $s'$.

## C.3. Reward Normalization

Reward normalization is a crucial component of our representation learning approach, as it relies directly on the reward function to guide the extraction and learning of learning. In experimental tasks, the rewards used for representation learning are consistent with those used in policy learning. Following Keypoint3D (Chen et al., 2021), to ensure stable learning dynamics, we apply a moving average normalization method to dynamically normalize the reward values. This method calculates the moving average of historical rewards and adjusts the rewards to have a mean of 0 and a standard deviation of 1. This normalization process helps mitigate fluctuations in reward values caused by variations in task difficulty, environmental changes, or exploration strategies, enabling the model to more effectively learn meaningful representations from stable

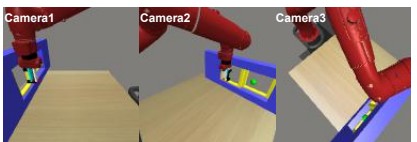 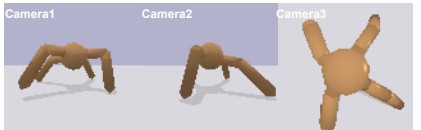 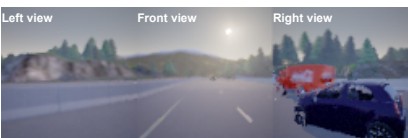

Figure 8: Visualization of three view observations in the Meta-World, PyBullet'Ant and CARLA environment. **Left & Middle:** Third-person view observations from three different cameras in the *'Close Window'* and PyBullet's Ant task. **Right:** First-person observations from three different views in the CARLA. It can be observed that in the Meta-World and PyBullet environments, each view exhibits a relatively high degree of coupling, encapsulating a substantial portion of the task-relevant information. However, due to factors such as occlusions (e.g., the robotic arm obstructing the target object or the leg joints in the PyBullet's Ant task), no single view captures the complete state information. In contrast, in the CARLA environment, the information contained in each view is mutually independent, and the three views collectively constitute the environment's state.

reward signals. Additionally, since the scale of rewards influences the bisimulation metric and the upper bound of value function errors, we adopt reward scaling to avoid feature collapse and reduce bisimulation measurement errors. Following the work of Zang et al. (2023), we apply $c_r = 1 - \gamma$ and $c_t = \gamma$ to compute bisimularity.

## D. Environment Details

**Meta-World.** To evaluate whether our model can accelerate policy optimization when jointly trained with the policy, we perform five complex robotic arm manipulation tasks in the Meta-World environment (Yu et al., 2020). Each task involves 50 randomized configurations, such as the initial pose of the robot, object locations, and target positions. As shown in Fig.8, for each task, we use three third-person cameras from different angles to observe the robot arm and relevant objects. Since the gripper state at the end of the robotic arm may not be clearly visible from any of the three camera angles, following the settings of Chen et al. (2021) and Hwang et al. (2023), an indicator is introduced in the Meta-World tasks to signify whether the gripper is open or closed. This indicator is concatenated with the learned latent state and is fed into the actor network. Due to experimental variations, we adopt the results reported in the F2C paper for comparison.

**PyBullet's Ant.** The PyBullet's Ant (Coumans & Bai, 2022) task is designed to simulate the motion control of a quadruped robot in a highly dynamic 3D-locomotion environment. The objective of this task is to control the joints of the robot's legs, enabling it to learn how to balance and move as quickly and stably as possible. The Ant robot has a highly dimensional state and action space, which includes physical quantities such as joint angles, angular velocities, and linear velocities. The robot's movement is generated by controlling the torque or force applied to its joints, making a fine-grained understanding of the movable joints and parts essential. As locomotion environments require temporal reasoning, we use a frame stack of 2. The reward function in this task is typically based on the robot's forward velocity while accounting for control costs (energy consumption) to incentivize efficient movement. Due to the complexity of the environment and the high-dimensional action space, the Ant task provides a significant challenge for training and testing RL algorithms.

**CARLA.** To evaluate the performance of our approach in more realistic multi-view tasks, we applied MFSC to the vision-based autonomous driving domain based on CARLA (Dosovitskiy et al., 2017). In this experiment, we select the official `Town04` map, featuring a circular highway with an intersection, as illustrated in Fig.9. The agent's goal is to drive as safely as possible for a limited 1,000-time steps, avoiding collisions with other moving vehicles or obstacles. Following the experimental setup of DBC (Zhang et al., 2021), the reward function is designed as $r_t(s, a) = \mathrm{v}^{\mathrm{T}} \hat{u}_{highway} \cdot \Delta t - \lambda_i \cdot impulse - \lambda_s \cdot |steer|$, where $\mathrm{v}^{\mathrm{T}}$ represents the velocity vector of the ego vehicle projected onto the unit vector of the highway $\hat{u}_{highway}$, and

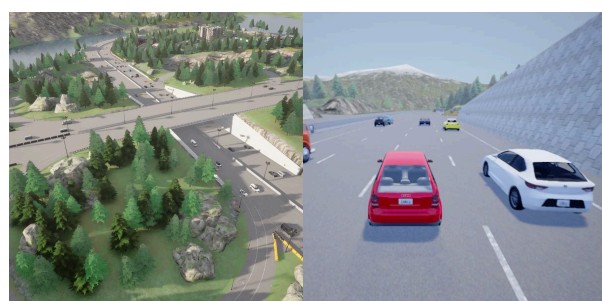

Figure 9: The bird's-eye view of the `Town04` map and snapshots of training period.

then is multiplied by time discretization $\Delta t$ to represent the effective distance. *Impulse* is the clash force, measured in Newton seconds, and *Steer* is the output of the steering amplitude. As shown in Fig.8, we use three cameras on the vehicle's roof, capturing left, front, and right views of the driver, each with a 60-degree field of vision. The agent must operate the vehicle's movement through first-person vision under varying conditions of wind, rain, clouds, or sunlight.

**DeepMind Control Suite.** The DeepMind Control Suite (DMControl) (Tassa et al., 2018) is a widely used benchmark environment for evaluating RL algorithms, offering a collection of continuous control tasks with interpretable rewards and standardized structures. Although DMControl is not inherently designed for multi-view inputs, we adapt it by utilizing historical time-series data as a proxy to simulate multi-view observations. This adaptation allows us to explore how temporal and multi-view information can enhance policy optimization. In our experiments, we select representative tasks from DMControl, including balance control, robotic manipulation, and motion planning, covering a range of complexities from simple to moderately challenging continuous control scenarios.

Multi-view observations in the Meta-World and PyBullet environments exhibit a greater degree of redundancy. This redundancy allows the attention mechanism to effectively focus on shared, useful information across views. However, in CARLA, although multiple views are provided, the differences between views can be much more pronounced. As shown in the right part of Fig.8, observations from different views are highly variable (e.g., different roads, and moving vehicles), leading to visual and semantic dissimilarity between different views. The larger differences between views make it harder to fuse the information effectively, and relying solely on attention mechanisms may not capture sufficient cross-view correlations. Therefore, concatenating image observations from multiple views is a straightforward yet effective approach.

# E. Additional Experimental Results

## E.1. Results on DeepMind Control Suite

To compare our method with the latest single-view RL methods, we downgrade MFSC to a single-view setting on the Deep-Mind Control Suite. Specifically, as shown in Fig.10, we construct "multi-view" observations by selecting observations $O_{t-3}, O_{t-2}, O_{t-1}, O_t$ from four consecutive time steps.

We selected the previously high-performing model-free RL algorithms SAC (Haarnoja et al., 2018), CURL (Laskin et al., 2020b), DrQ-v2 (Yarats et al., 2022) and TACO (Zheng et al., 2023), along with the model-based RL algorithms Dreamer-v3

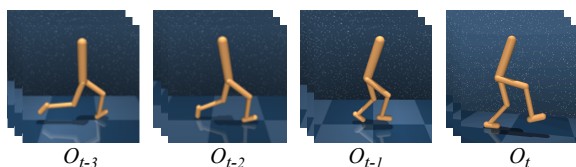

Figure 10: "Multi-view" observation in DMControl. Each observation consists of a stack of three consecutive image frames.

(Hafner et al., 2023) and TD-MPC2 (Hansen et al., 2024). The results of DrQ-v2, Dreamer-v3 and TD-MPC2 are obtained from Yarats et al. (2022), Hafner et al. (2023), Hansen et al. (2024). The reproduced results for TACO are implemented by the official code [1]. The results of SAC and CURL are downloaded from this link [2].

The results, as summarized in Table 4, indicate that MFSC, when degraded to a single-view setting, performs competitively with established single-view RL algorithms. Although there is a slight performance gap compared to the SOTA algorithm TACO, MFSC consistently achieves results on par with first-tier methods such as Dreamer-v3 and TD-MPC2 across the evaluated tasks, and it surpasses that of SAC, CURL, and DrQ-v2 across most tasks. Furthermore, these results demonstrate the efficacy of MFSC's representation learning framework, even when reduced to a sequential single-view scenario. Despite the constrained single-view conditions, MFSC's ability to remain competitive reaffirms the robustness of its design and its suitability for a wide range of RL tasks.

Table 4: Comparison results on DeepMind Control Suite. The results of MFSC and TACO are presented as the mean and standard deviation across four random seed runs. The best score is marked with **colorbox**.

|  | Model-free RL | | | | | Model-based RL | |
| --- | --- | --- | --- | --- | --- | --- | --- |
| Task (1M Steps) | MFSC(ours) | SAC | CURL | DrQ-v2 | TACO | Dreamer-v3 | TD-MPC2 |
| Cartpole Swingup | 832±29 | 717±134 | 758±155 | **865±15** | 864±2 | 862±9 | − |
| Cheetah Run | 679±33 | 19±14 | 502±120 | 710±52 | **915±6** | 841±87 | 537±60 |
| Finger Spin | 977±7 | 289±104 | 873±145 | 860±104 | 978±10 | 594±136 | **985±2** |
| Walker walk | 909±40 | 41±17 | 818±261 | 770±371 | **968±3** | 959±27 | 940±8 |

---

[1] https://github.com/FrankZheng2022/TACO
[2] https://github.com/danijar/dreamerv3/tree/main/scores

## E.2. Results on CARLA

The results of CARLA are shown in Table 5. MFSC achieves the highest performance on three out of six driving metrics. Our method outperforms SAC on both crucial comparisons of episodic reward and episodic distance. Furthermore, in terms of metrics reflecting vehicular stability during operation, such as average steering amplitude variation, average braking intensity, and collision severity, all algorithms exhibited relatively modest performance. This is attributed to the reward function being primarily crafted to validate representation learning rather than specifically tailored to address safety considerations. Overall, in the CARLA task, our algorithm does not exhibit a significant improvement over the baseline algorithm SAC. This further supports our previous viewpoint that, in multi-view tasks like CARLA, where there is almost no shared information between views, concatenating image observations from multiple views, as in SAC, is a straightforward yet effective approach.

Table 5: Comparison results of different driving metrics in CARLA. Results of DrQ (Yarats et al., 2021b) and CURL (Laskin et al., 2020b) are obtained from Liang et al. (2024). The best score is marked with **colorbox**. $\pm$ corresponds to a standard deviation of three random seeds, after 400k training steps. The direction of the arrow indicates whether the desired metric is large or small.

| Metrics | SAC | DrQ | CURL | MFSC(ours) |
|---|---|---|---|---|
| Distance (m) $\uparrow$ | $179.60 \pm 76.29$ | $158.10 \pm 1.54$ | $172.50 \pm 7.60$ | $\mathbf{183.30} \pm 67.39$ |
| Successes episodes $\uparrow$ | **21%** | 21% | 14% | 13% |
| Average steer $\downarrow$ | 10.24% | 12.02% | 9.76% | **8.94%** |
| Avergae brake $\downarrow$ | **0.94%** | 1.66% | 1.75% | 1.02% |
| Crash intensity (N/s) $\downarrow$ | 3750 | **3313** | 4311 | 4211 |
| Episode return $\uparrow$ | $152.73 \pm 4.54$ | $122.80 \pm 2.58$ | $149.10 \pm 14.10$ | $\mathbf{168.01} \pm 9.62$ |

## E.3. Visualization and Analysis on Noisy View

To evaluate the robustness of our method under interference views, we introduced a noisy-views setting. Specifically, as illustrated in Fig.11, in addition to the clean views $O_1, O_2, O_3$, that are normally acquired, we incorporate an additional viewpoint $O_4$ that is entirely irrelevant to the task. The noisy view $O_4$ consists of distracting images, all sourced from the `video_hard` datasets in *DMControl Generalization Benchmark* (DMControl-GB) [3]. We employed the same Grad-CAM technique as in Fig.4

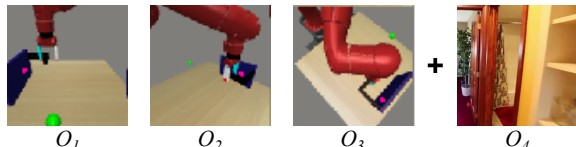

Figure 11: Multi-view observations in noisy view setting.

to visualize the features of the four views. As shown in the visualization results of Fig.12, our method remains capable of capturing task-relevant details even under the noisy interference setting.

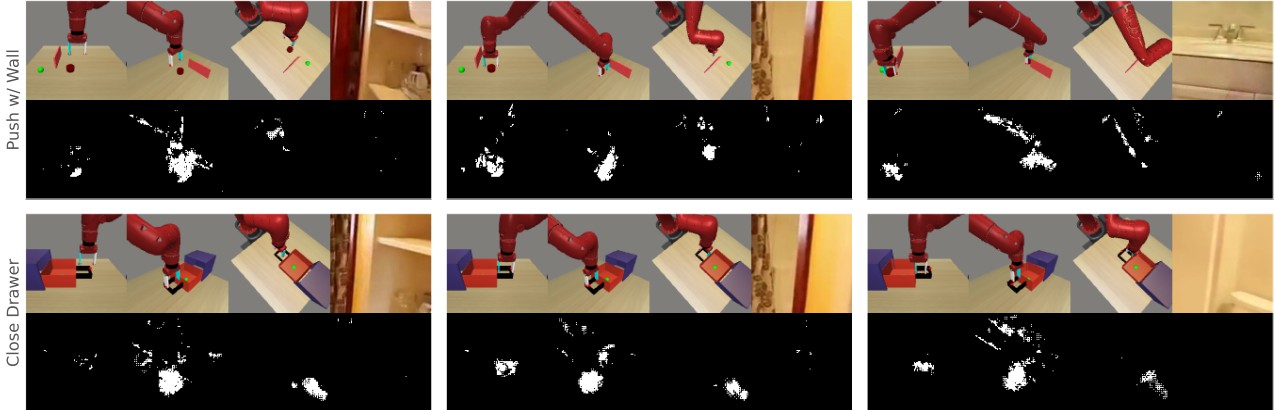

Figure 12: Visualization of learned representation on noisy view. Pixels with the highest gradient values (top 2.5%) across all views are marked white while others are marked black.

---

[3] https://github.com/nicklashansen/dmcontrol-generalization-benchmark

To investigate whether bisimulation can assist MFSC in countering the interference of noise, we conducted ablation experiments under the setting of interference views. The results of Fig.13(a) confirm that bisimulation effectively aids MFSC in mitigating the impact of noise, allowing it to maintain its performance even in the presence of disruptive noise. This demonstrates the robustness of MFSC when enhanced with bisimulation, highlighting its ability to focus on relevant features while disregarding irrelevant noise.

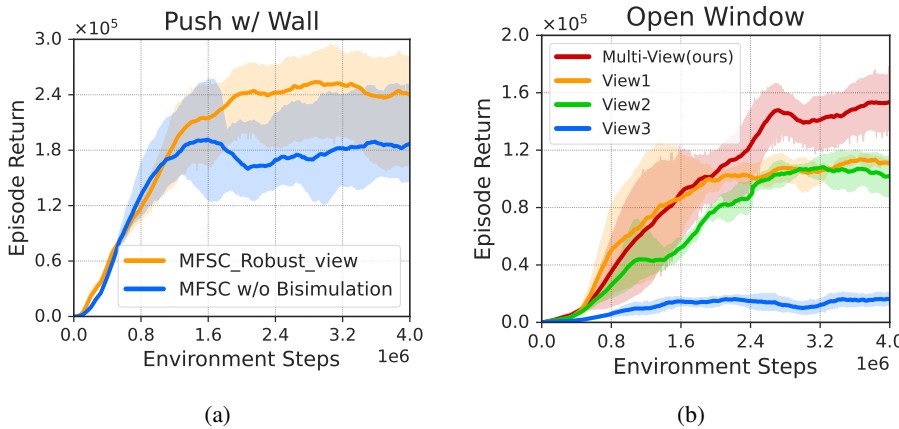

Figure 13: (a) Ablation study on robust view. (b) Ablation study on varying view counts.

### E.4. Ablation Study on Varying View Counts

We have added an ablation study that varies the number of views, and the results are presented in Fig13(b). It shows that using any single view out of the three performs worse than the full multi-view setup. In particular, View 3 — a top-down perspective — suffers from severe occlusion, resulting in the lowest performance among the three. This highlights the limitations of relying on individual views and underscores the effectiveness of our approach in extracting and fusing complementary information across views. Overall, these results validate the necessity and benefit of multi-view learning in achieving more robust and informative state representations.

### E.5. Quantitative Analysis on Bisimulation Metric

To provide a more quantitative analysis of the bisimulation metric, we include the training curve of the bisimulation loss and additionally measure the mutual information $I\left(z_L^0; s\right)$ between the final fused embeddings $z_L^0$ and the ground-truth states $s$ using the MINE[4] method. As illustrated in Fig.14, with the progression of training and the convergence of bisimulation loss, we observe a consistent increase in mutual information. This indicates that the model not only optimizes the bisimulation criterion but also gradually constructs a task-relevant representation space aligned with the true environment states.

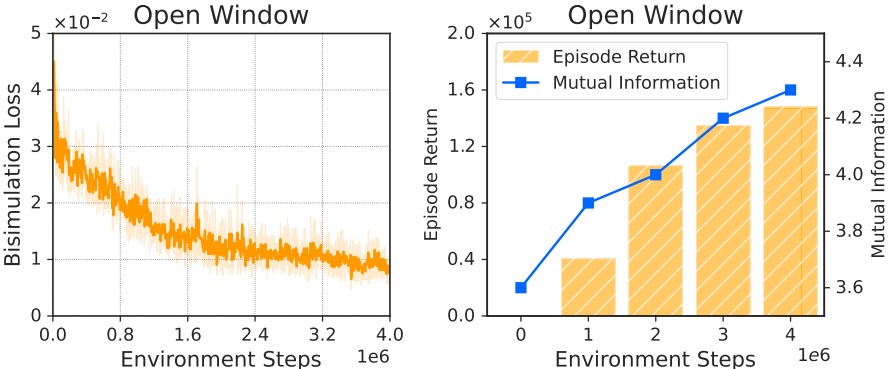

Figure 14: Quantitative analysis on bisimulation metric.

---

[4]https://github.com/gtegner/mine-pytorch

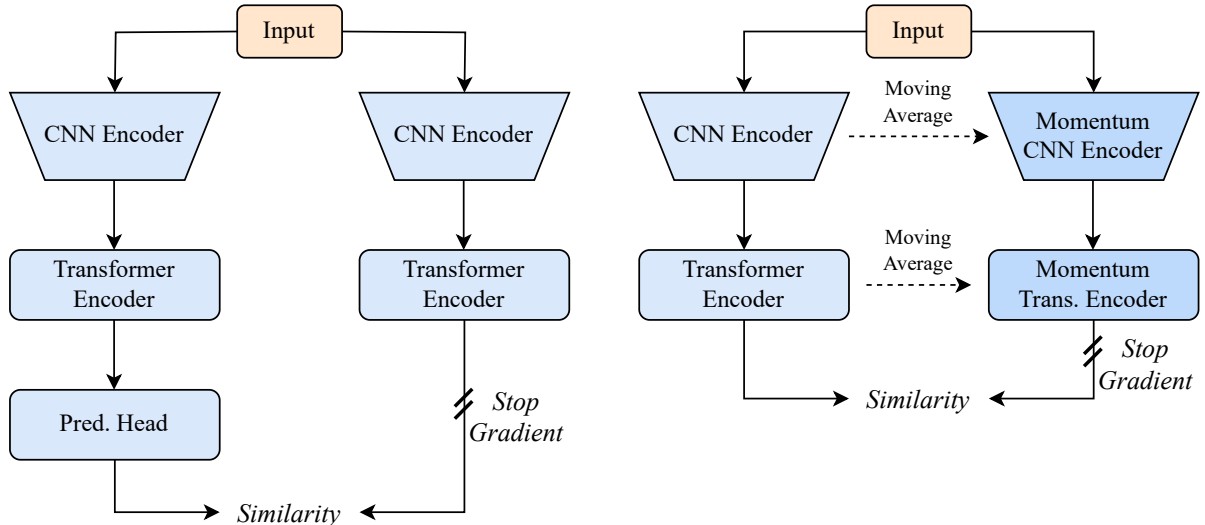

Figure 16: Comparison on network architectures. **Left:** SimSiam architecture. **Right:** Momentum encoder architecture. **'//'** represents a stop-gradient operation.

### E.6. Hyperparameter Sensitivity Analysis

In the overall loss function of MFSC in Eq.10, the weighting coefficient $\lambda$ is set to 1, signifying that both loss components contribute equally to the total loss. We also conducted a hyperparameter sensitivity analysis to evaluate the impact of different fusion loss and reconstruction loss contributions on the algorithm's performance. Specifically, we compared three configurations: $\lambda = 1$ (original setting), indicating equal contributions from both losses, $\lambda = 2$, signifying a greater contribution from the fusion loss, and $\lambda = 0.5$, denoting a greater influence from the reconstruction loss. The results are visualized in Fig.15 for two tasks: *'Close Box'* and *'Ant No Color'*.

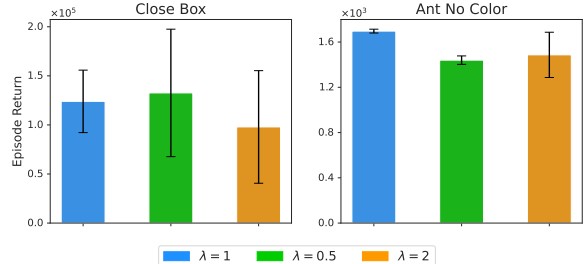

Figure 15: Results of loss weight tuning experiment. The vertical lines on the barchart represent the standard deviation of the results from the three seeds.

The experimental results indicate that the model performs relatively better when the weights of the two losses are equal (i.e., $\lambda = 1$). Increasing or decreasing the weight of either loss may result in performance variations, likely due to the model overemphasizing one loss component while neglecting the importance of the other. This finding underscores the critical importance of balancing the fusion loss and the reconstruction loss to optimize both the overall performance and stability of the model.

### E.7. Results on Different Network Architectures

To verify the robustness of our method in different network architectures, we implemented two distinct MFSC network designs and performed evaluations in the Meta-World environment. As illustrated in the left part of Fig.16, the network employed in the paper adopts a structure which is similar to the SimSiam (Chen & He, 2021) architecture. The encoder includes all layers that can be shared between both branches. To avoid collapsing, we applied a stop-gradient operation to one branch of the network. Furthermore, we also implemented MFSC with a network architecture based on a momentum encoder. As shown in the right half of Fig.16, in addition to the stop-gradient operation, the momentum encoder is updated using the exponential moving average method.

In Fig.17, it is evident that although the performance of the MFSC with two distinct network architectures differs on certain tasks, such as *'Open Window'* and *'Push w/Wall'*, overall, the algorithms exhibit comparable performance across both architectures. This outcome suggests that our method demonstrates a certain degree of robustness with respect to the design of the network architecture.

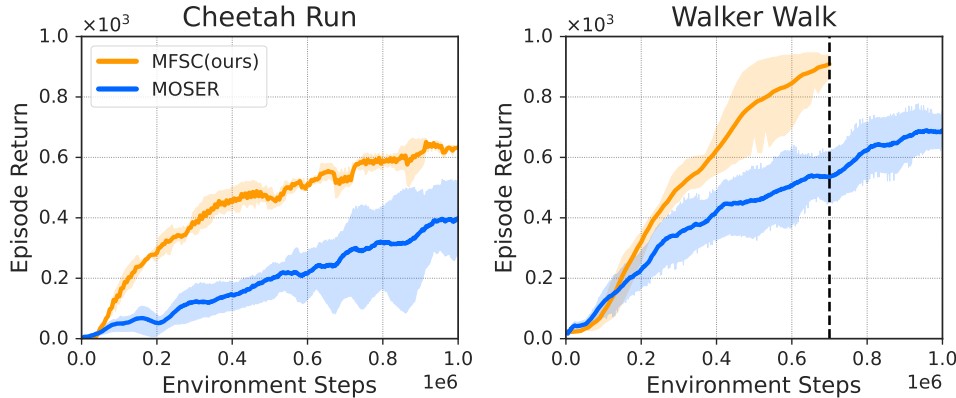

Figure 17: Performance comparison of two different network architectures on Meta-World.

### E.8. Results of the Comparison with Additional MVRL Algorithm

We compare our method with the model-based MVRL algorithm MOSER (Wan et al., 2024). MOSER seeks the optimal viewpoint for learning task representations under multiple views. Its implementation code is available at this link[5]. The corresponding experimental results are illustrated in Fig.18. Our experimental results show that our approach outperforms the MOSER algorithm in both tasks, demonstrating the effectiveness of our method.

Figure 18: Performance comparison of an MB-MVRL algorithm MOSER.

### E.9. Comparison of Training Resources

We compared the training time, inference time, and resource requirements of MFSC with the following baselines: single-view methods RAD and Vanilla-PPO, as well as multi-view methods Keypoint3D and LookCloser. The experiments were conducted on a server equipped with two NVIDIA A800 GPUs (Ampere architecture) with 80GB of VRAM each, running on CUDA 12.3 and NVIDIA driver version 545.23.06. The server is powered by dual Intel Xeon Platinum 8370C CPUs, each featuring 32 cores and 64 threads, with a base clock speed of 2.80GHz and a maximum clock speed of 3.5GHz. The system has a total of 128 logical processors and a NUMA configuration with two nodes. The hardware setup ensures sufficient computational resources to train and evaluate the proposed method.

Table 6: Comparison of training resources for different algorithms. The last column represents the improvement in the average score of five tasks in Meta-World compared to the baseline algorithm RAD. Optimal results in the MVRL algorithms are marked as bold and underlined.

| Category | Method | Training time | Inference time | GPU memory | Performance improvement |
|---|---|---|---|---|---|
| View Concatenating | Vanilla PPO | 3.8h | 2.67ms | 850MB | — |
| | RAD | 3.8h | 2.69ms | 758MB | — |
| View Fusion | Keypoint3D | **12.2h** | 6.01ms | **1406MB** | 21.17% ↑ |
| | LookCloser | 13.9h | 14.35ms | 3530MB | 45.66% ↑ |
| | **MFSC (ours)** | 14.9h | **4.26ms** | 4098MB | **97.93%** ↑ |

---

[5] https://github.com/yixiaoshenghua/MOSER

As shown in Table 6, due to the complexities of information fusion, model architecture, and cross-view feature extraction, multi-view fusion algorithms typically require more resources than simpler multi-view image concatenating methods. Compared to other multi-view fusion algorithms, our method demands greater training time and GPU memory. However, fortunately, our algorithm significantly outperforms others in terms of inference time and performance improvement. This indicates that while our method may require more resources during the training phase, it offers much more efficient performance during the inference phase, which is a crucial advantage in practical applications.

## F. Algorithm

Our pseudocode in Algorithm 1 and Algorithm 2, where Algorithm 1 is based on Proximal Policy Optimization (PPO) and Algorithm 2 is based on Soft Actor-Critic (SAC). Furthermore, we illustrate the representation learning component of MFSC in Algorithm 3 using a PyTorch-like style for enhanced clarity and comprehension.

---

**Algorithm 1** Pseudocode of MFSC, based on PPO

---

1: **Input:** iterations to repeat entire processes $N_{\text{Repeat}}$, batch size $B$, rollout length $T$.
2: **for** iter $= 1$ **to** $N_{\text{Repeat}}$ **do**
3:     **Initialize** $\mathcal{B}_{rollout}$
4:     **for** $b = 1$ **to** $B$ **do**
5:         Run policy $\pi_{\theta_{old}}$ to collect $(\vec{o}, a, r, \vec{o}')_{1:T}$
6:         $\mathcal{B}_{rollout} \leftarrow \mathcal{B}_{rollout} \cup (\vec{o}, a, r, \vec{o}')_{1:T}$
7:     **end for**
8:     Estimate advantage values $\hat{A}_{1:T, 1:N}$ on $\mathcal{B}_{rollout}$
9:     **for** $t = 1$ **to** $T$ **do**
10:        Sample $(\vec{o}, a, r, \vec{o}') \sim \mathcal{B}_{rollout}$
11:        Patch masking the multi-view observation $\vec{o}$
12:        Calculate $\mathcal{L}_{policy}$   $\triangleright$ PPO algorithm
13:        Calculate $\mathcal{L}_{fus}$ according to Eq.8
14:        Calculate $\mathcal{L}_{rec}$ according to Eq.9
15:        Calculate $\mathcal{L}_{dyn}$ according to Eq.13
16:        Optimize $\mathcal{L}_{policy} + \mathcal{L}_{rec} + \mathcal{L}_{fus} + \mathcal{L}_{dyn}$ throughout $\mathcal{B}_{rollout}$
17:     **end for**
18:     $\pi_{\text{old}} \leftarrow \pi$
19: **end for**

---

---

**Algorithm 2** Pseudocode of MFSC, based on SAC

---

1: **Input:** maximum time steps $T$, batch size $B$, replay buffer size $|\mathcal{D}|$, target network update frequency $\tau$.
2: **for** Time $t = 0$ to $T$ **do**
3:     Encode state: $\mathbf{z_t} = \phi(\tilde{\mathbf{o}})$
4:     Execute action: $\mathbf{a_t} \sim \pi(\mathbf{z_t})$
5:     Record data: $\mathcal{D} \leftarrow \mathcal{D} \cup \{\vec{o}_t, a_t, r_t, \vec{o}'_{t+1}\}$
6:     Sample batch: $B_t \sim \mathcal{D}$
7:     Update value network: $\mathbb{E}_{\mathcal{B}_t}\big[\mathcal{L}(V)\big]$   $\triangleright$ SAC algorithm
8:     Update policy network: $\mathbb{E}_{\mathcal{B}_t}\big[\mathcal{L}(\pi)\big]$   $\triangleright$ SAC algorithm
9:     Update fusion network $\phi$: $\mathbb{E}_{\mathcal{B}_t}\big[\mathcal{L}(\phi)\big]$ according to Eq.10
10:    Update the dynamics: $\mathbb{E}_{B_i}[\mathcal{L}_\mathcal{P}(\theta_j)]$, where $j \in \{1, \ldots, K\}$ according to Eq.13
11: **end for**

---

---

**Algorithm 3** Pseudocode for the representation learning component of MFSC, PyTorch-like

---

```python
# encoder: cnn + mlp + self-attention, fusion network, the output is L2-normalized
# transition_model: mlps, ensemble of transition models, the output is L2-normalized
# global_predictor: mlp, prediction head network
# lmb: weighting coefficient

def cosine_dis(feature_a, feature_b):
    feature_a_norm = nn.functional.normalize(feature_a, dim=1)
    feature_b_norm = nn.functional.normalize(feature_b, dim=1)
    temp = torch.matmul(feature_a_norm, feature_b_norm.T)
    dis = torch.diag(temp)
    return dis

def spr_loss(masked_z_a, target_masked_z_a):
    global_latents = self.global_predictor(masked_z_a)
    with torch.no_grad():
        global_targets = target_masked_z_a
    loss = norm_mse_loss(global_latents, global_targets, mean=False).mean()
    return loss

def compute_mfsc_loss(obs, action, reward, next_obs):

    obs = obs.view(batch_size, self.num_cameras, self.frame_stack, self.image_channels, self.image_size, self.
        image_size)
    next_obs = next_obs.view(batch_size, self.num_cameras, self.frame_stack * self.image_channels, self.
        image_size, self.image_size)

    # reconstruction loss
    masked_z_a = self.encoder.mask_encode(obs, mask=True)
    with torch.no_grad():
        target_masked_z_a = self.encoder.mask_encode(obs, mask=False)

    masked_z_a = masked_z_a.flatten(0, 1)
    target_masked_z_a = target_masked_z_a.flatten(0, 1)

    res_loss = spr_loss(masked_z_a, target_masked_z_a)

    # fusion loss
    z_a = self.encoder(obs)
    with torch.no_grad():
        pred_a = self.transition_model.sample_prediction(torch.cat([z_a, action], dim=1))

    next_diff = compute_dis(pred_a, pred_a)
    r_diff = torch.abs(reward.T - reward)
    z_diff = compute_dis(z_a, z_a)

    bisimilarity = self.c * r_diff + (1 - self.c) * next_diff
    fus_loss = torch.nn.HuberLoss()(z_diff, bisimilarity.detach())

    # mfsc loss
    mfsc_loss = fus_loss + self.lmb * res_loss

    # transition loss
    h = self.encoder(obs)
    next_h = self.encoder(next_obs).unsqueeze(0)
    pred_next_h = self.transition_model(torch.cat([h, action], dim=1))

    cos_sim = F.cosine_similarity(next_h.detach(), pred_next_h, dim=-1)
    transition_loss = 1 - cos_sim.mean()

    return mfsc_loss, transition_loss
```

---

