# OpenReview forum: "Learning Fused State Representations for Control from Multi-View Observations"
_ICML.cc/2025/Conference — ICML 2025 poster_

### Official Review · Reviewer_NPK4 · 2025-02-25

**Overall Recommendation:** 4

**Summary:**

The paper proposes a novel method called Multi-view Fusion State for Control (MFSC) to improve Multi-View Reinforcement Learning (MVRL). MFSC's main contribution is its integration of bisimulation metric learning into the MVRL framework, allowing for the extraction of task-relevant representations from multi-view observations. Additionally, the authors introduce a multiview-based masking and latent reconstruction auxiliary task to enhance the robustness of representation learning, notably when views are missing or noisy. The MFSC framework utilizes a Self-Attention Fusion Module that fuses multi-view representations using a learnable state fusion embedding. The bisimulation metric guides the learning of task-relevant features, while the auxiliary masking task helps the model deal with missing views by learning shared representations across different views. Key results show that MFSC outperforms existing MVRL methods in several tasks, including robotic manipulation and locomotion tasks, demonstrating superior performance in environments with missing or noisy views. The method is remarkably robust to interference and can handle incomplete observations effectively. The paper also includes visualizations and ablation studies that validate the effectiveness of each component, such as the bisimulation constraint and the self-attention mechanism, in improving task performance.

**Claims And Evidence:**

Yes, the paper's claims are generally supported by clear and convincing evidence. However, one potential area for further clarification is the handling of complex scenarios with significant occlusions or highly incomplete views. While the method shows robustness in many cases, some views may still contain critical information that is difficult to reconstruct, especially in tasks involving complex state transitions. This limitation could affect the performance in very challenging real-world environments, and further research into addressing these edge cases would strengthen the overall claims.

**Essential References Not Discussed:**

Multiview Dreaming[1] extends the Dreaming algorithm to achieve integrated recognition and control from multiple perspective observations. This method utilizes contrastive learning to train a shared latent space between different views and integrates the latent state distributions from multiple perspectives through an expert mixture approach, thereby addressing the limitations of single-view observations in traditional reinforcement learning methods. MOSER[2] is a model-based approach that actively seeks the optimal perspective for learning task representations under multiple views to enhance performance.

**References**
[1] Kinose, A., Okada, M., Okumura, R., & Taniguchi, T. (2023). Multi-view dreaming: Multi-view world model with contrastive learning. Advanced Robotics, 37(19), 1212-1220.
[2] Wan, S., Sun, H. H., Gan, L., & Zhan, D. C. (2024, August). MOSER: learning sensory policy for task-specific viewpoint via view-conditional world model. In Proceedings of the Thirty-Third International Joint Conference on Artificial Intelligence (pp. 5046-5054).

**Experimental Designs Or Analyses:**

**Baselines and Comparisons**

**Strengths:**

The authors compared MFSC with several baseline methods, including Keypoint3D, LookCloser, F2C, MVD, RAD, and Vanilla PPO. These baselines cover a range of approaches to MVRL, providing a robust comparison. The results show that MFSC consistently outperforms these baselines in most tasks, demonstrating the effectiveness of the proposed method.

**Potential Issues:**

Missing Baselines: As mentioned earlier, the comparisons do not include some recent MBMVRL methods. Including these methods would strengthen the validity of the results.

Hyperparameter Tuning: The paper does not provide detailed information on the hyperparameter tuning for the baselines. If the baselines were not tuned to their optimal settings, the comparisons might be unfair.

**Performance Metrics and Evaluation**

**Strengths:**

The authors evaluated the performance of their method using various performance metrics, including episode return, environment steps, and success rates. Using multiple seeds and confidence intervals in the results helps assess the statistical significance of the findings.

**Potential Issues:**

Limited Metrics in CARLA: In the CARLA environment, the evaluation metrics primarily focus on driving performance (e.g., distance traveled, success rate, steering amplitude, braking intensity, and collision severity). Including additional metrics, such as the number of collisions or the time taken to complete the task, could provide a more comprehensive evaluation.

Reward Normalization: The paper uses reward normalization to stabilize the learning process. While this is a common practice, it could mask some of the nuances in the reward signal. The authors should discuss the impact of reward normalization on the results.

**Methods And Evaluation Criteria:**

The comparison methods are all model-free multiview reinforcement learning approaches, lacking a comparison with model-based methods, which makes it difficult to fully convince us of the performance superiority of this method.

**Other Comments Or Suggestions:**

Please see the Questions.

**Other Strengths And Weaknesses:**

**Strengths**

- The paper presents a novel approach integrating bisimulation metric learning with a multi-view-based masking and latent reconstruction auxiliary task.
- The paper is well-structured and clearly presents the methodology, experiments, and results.
- The authors conducted extensive experiments to validate the effectiveness of MFSC.

**Weaknesses**

- The lack of comparison with model-based methods makes it impossible to fully validate the superiority of this approach.
- There is a lack of quantitative analysis and presentation regarding the Bisimulation metric, not just ablation experiments.

**Questions For Authors:**

1. Could you provide more comparisons with model-based methods, such as MV-MWM, Multiview Dreaming, and MOSER?
2. Will conducting reinforcement learning from multiple perspectives lead to increased time and storage costs, and how can this issue be mitigated?

**Relation To Broader Scientific Literature:**

Bisimulation Metric Learning: Extends the concept of bisimulation metrics to the multi-view setting, providing a novel approach to learning task-relevant representations.

Application to Real-World Scenarios: Evaluates the method in a real-world autonomous driving environment, highlighting its potential for practical applications.

**Theoretical Claims:**

Yes, the proof of Value Function Bound is convincing.

---

> ### Author Rebuttal · Authors · 2025-03-31
>
> Thanks for your comments, and we will address your concerns in the following.
>
> ------
>
> **Potential Issues: hyperparameter tuning**
>
> Thank you for your valuable feedback. For all baseline methods with publicly available implementations, we directly used their officially released code and default configurations to ensure they achieve the performance reported in the original papers.  We will include a clarification of this in the experimental details section.
>
> ------
>
> **Potential Issues: limited metrics in CARLA**
>
> Thank you for your suggestion. Our choice of evaluation metrics in CARLA is consistent with previous works. In addition, we would like to clarify that our evaluation metrics not only focus on driving performance (e.g., distance traveled, success rate of reaching 100 meters), but also take driving safety into account, including steering amplitude, braking intensity, and collision severity.
>
> ------
>
> **Potential Issues: reward normalization**
>
> We acknowledge the reviewer’s point that reward normalization may mask certain nuances in the reward signal. In our experiments, whether to apply reward normalization under different benchmarks depends on the implementation of the methods we compare against. We also conducted ablation studies on MetaWorld tasks, where reward normalization is required, to analyze the impact of reward normalization. Detailed results can be found at this **[link](https://anonymous.4open.science/r/MFSC-1488/rebuttal.pdf)** (Fig.2). The results show that without reward normalization, the performance of our method degrades significantly. This is because the large scale of raw rewards causes large gradients during Q-value updates, leading to oscillations in Q-value loss and overall instability in training. Additionally, reward normalization constrains the reward signal to a feasible range, which helps stabilize the optimization of the bisimulation objective, further improving the quality of the learned representations.
>
> ------
>
> **Essential References Not Discussed**
>
> We appreciate the additional references provided and are pleased to include Multi-view Dreaming and MOSER in our manuscript, as they significantly enhance the completeness and relevance of our work.
>
> ------
>
> **W1 and Q1: the lack of comparison with model-based MVRL**
>
> We appreciate the reviewer’s suggestion to compare our method with MB-MVRL algorithms. Since MV-MWM relies on additional expert demonstrations and Multiview Dreaming does not release its implementation code, we chose to compare our method with the MB-MVRL algorithm MOSER. MOSER seeks the optimal viewpoint for learning task representations under multiple views. Its implementation code is available at https://github.com/yixiaoshenghua/MOSER, and the corresponding experimental results can be found at this **[link](https://anonymous.4open.science/r/MFSC-1488/rebuttal.pdf)** (Fig.3). Our experimental results show that our approach outperforms the MOSER algorithm in both tasks, demonstrating the effectiveness of our method. We believe these additional comparisons help validate the superiority of our method.
>
> ------
>
> **W2: quantitative analysis on bisimulation metric**
>
>  Thanks for your insightful comment. To provide a more quantitative analysis of the bisimulation metric, we include the training curve of the bisimulation loss and additionally measure the mutual $I\left( z_{L}^{0};s \right)$ information between the final fused embeddings $z_{L}^{0}$ and the ground-truth states $s$ using the MINE [1] method. The results can be found at this **[link](https://anonymous.4open.science/r/MFSC-1488/rebuttal.pdf)** (Fig.4). As training progresses and the bisimulation loss converges, we observe a consistent increase in mutual information. This indicates that the model not only optimizes the bisimulation criterion but also gradually constructs a task-relevant representation space aligned with the true environment states.
>
> [1]Belghazi, et al. Mutual information neural estimation.
>
> ------
>
> **Q2: Will conducting reinforcement learning from multiple perspectives lead to increased time and storage costs, and how can this issue be mitigated?**
>
> While multi-view observations provide a more comprehensive understanding of the environment, they may lead to increased computational time, storage cost. To mitigate this, we can leverage pretraining techniques to reduce training costs. Additionally, inspired by MOSER, we can train a view selector to dynamically choose the most relevant views for decision-making, thus may potentially minimize unnecessary computational and storage demands.
>
> ------
>
> Please do not hesitate to let us know if you have any additional comments.

---

> > ### Comment · Reviewer_NPK4 · 2025-04-08
> >
> > Thanks for the author's reply. My problems have been solved, and I will raise the score from 3 to 4.

---

### Official Review · Reviewer_uqci · 2025-03-10

**Overall Recommendation:** 4

**Summary:**

The paper proposes a novel framework for Multi-View Reinforcement Learning (MVRL). The key contributions include:

- Integrating bisimulation metric learning into MVRL to extract task-relevant representations from multi-view observations.

- Introducing a multiview-based masking and latent reconstruction auxiliary task to enhance robustness against missing or noisy views.

- Demonstrating superior performance over existing methods in robotic manipulation (Meta-World, PyBullet) and autonomous driving (CARLA) tasks, especially in scenarios with interference or missing views.

**Claims And Evidence:**

Yes, the claims are generally supported by clear and convincing evidence.

**Essential References Not Discussed:**

No, the essential references are discussed in the paper.

**Experimental Designs Or Analyses:**

The experimental designs and analyses are largely sound but exhibit minor validity concerns in specific benchmarks and theoretical assumptions. For example, on DeepMind Control Suite environment, temporal stacking conflates temporal and spatial multi-view learning, undermining conclusions about MFSC’s multi-view fusion capabilities.

**Methods And Evaluation Criteria:**

Yes. The methods are largely appropriate for addressing the challenges of multi-view reinforcement learning.

**Other Comments Or Suggestions:**

- The paper fixes the number of views to 3. How does the number of views impact the experimental results? An ablation study on varying view counts would strengthen the analysis. If performance with 1 view is comparable to multi-view setups, it raises questions about the necessity of multi-view learning or whether the method fails to extract useful cross-view information.

- In Fig.1, the symbol $\tilde{z}^n_L$ was mistakenly written as $\tilde{x}^n_L$.

**Other Strengths And Weaknesses:**

Strengths:

- This paper is clearly written and easy to follow.

- It introduces a novel framework by integrating bisimulation metrics with multi-view fusion and masking-based reconstruction, addressing critical gaps in task-relevance and robustness for MVRL.

- The work is supported by extensive experiments across diverse benchmarks (Meta-World, PyBullet, CARLA), including ablation studies and robustness tests under noisy/missing views.

Weaknesses：

- Using sequential frames as different views in the DeepMind Control environment is an invalid design choice. A more reasonable approach might involve training with three identical images as inputs to isolate multi-view fusion setting.

**Questions For Authors:**

In Fig.1, I would like to understand the model’s design details: How does the architecture ensure that $z_L^0$ the fused state embedding) integrates features from all views? Is this achieved by blocking gradient propagation from $z_L^{1-3}$, and is there theoretical justification for this design?

**Relation To Broader Scientific Literature:**

The paper advances MVRL by integrating bisimulation metrics and masking-driven reconstruction into a cohesive framework. MV-MWM (Seo et al., 2023a) relies on pixel-level reconstruction and expert demonstrations, which may retain redundant details. In this paper, the latent reconstruction loss avoids reconstructing irrelevant details, improving efficiency.

**Theoretical Claims:**

Yes. The theoretical claims are sound.

---

> ### Author Rebuttal · Authors · 2025-03-31
>
> We sincerely thank the reviewer for thoughtful and constructive feedback. We are pleased that our proposed framework for Multi-View Reinforcement Learning (MVRL), including the integration of bisimulation metric learning and masking-based latent reconstruction, was well-received. We appreciate the recognition of our theoretical formulation, experimental design, and the robustness of our method across diverse benchmarks. We truly appreciate your insightful feedback and constructive suggestions, and we will address your concerns in the following.
>
> ------
> **W1: experimental setup of Deep Mind Control Suite**
>
> We would like to clarify that our primary goal is to explore how multi-view information can be leveraged to enhance decision-making efficiency. While we focus on spatial multi-view in the paper, as demonstrated in our experiments with environments like Metaworld, PyBullet, and CARLA, we also recognize that "multi-view'' can manifest in both temporal and even multi-modal forms. Hence we further explore the temporal information fusion in DeepMind Control environments. To address the potential confusion, we will clarify the distinction between these two experimental setups in the next version of the paper.
>
> ------
>
> **Other Comments Or Suggestions 1: ablation study on varying view counts**
>
> We appreciate the reviewer’s insightful suggestion. In response, we have added an ablation study that varies the number of views, and the results are presented in Fig. 1 at the following link **[link](https://anonymous.4open.science/r/MFSC-1488/rebuttal.pdf)**. It shows that using any single view out of the three performs worse than the full multi-view setup. In particular, View 3 — a top-down perspective — suffers from severe occlusion, resulting in the lowest performance among the three. This highlights the limitations of relying on individual views and underscores the effectiveness of our approach in extracting and fusing complementary information across views. Overall, these results validate the necessity and benefit of multi-view learning in achieving more robust and informative state representations.
>
> ------
> **Other Comments Or Suggestions 2: In Fig.1, the symbol was mistakenly written.**
>
> Thanks for your careful review and for pointing out the mistake. We appreciate your feedback and will correct the symbol in the next version of the paper.
>
> ------
>
> **Q: model design details**
>
> Thanks for your comment, and we apologize for any confusion caused. We clarify this design as follows: we introduce an additional learnable ``[state]`` embedding that interacts with the view-specific embeddings through a self-attention mechanism. The output of this embedding, denoted as $z_L^{0}$, is updated through the bisimulation loss. The theory of bisimulation guarantees that the learned representations retain all task-relevant information from multiple views, enabling $z_L^{0}$ to effectively aggregate this information into a unified representation. Importantly, MFSC optimizes the bisimulation objective exclusively with respect to $z_L^{0}$, not the view-specific embeddings $z_L^{1-3}$. This design choice follows a similar paradigm to BERT and ViT, where supervision is applied exclusively to a designated token (e.g., the ``[class]`` token). We also recognize that the current depiction of gradient flow in Fig.1 may be misleading, and we will revise it in future versions to accurately reflect the optimization process.
>
> ------
>
> Please do not hesitate to let us know if you have any additional comments.

---

> > ### Comment · Reviewer_uqci · 2025-04-07
> >
> > Thank you for the authors for the detailed response and additional results. I will maintain my score of 4 and recommend acceptance of this paper.

---

### Official Review · Reviewer_RYsq · 2025-03-14

**Overall Recommendation:** 4

**Summary:**

The paper proposes a novel framework for multi-view reinforcement learning to effectively learn task-relevant representations of the state from multi-view observations. The new framework not only incorporates the bisimulation metric which aligned representation with the task’s objectives but also add latent reconstruction as an auxiliary task to retain crucial details that are specific to each individual view. The paper also provide strong experimental results in realistic scenarios which demonstrate the effectiveness of the unified fusion state representation from multi-view setting.

**Claims And Evidence:**

The paper provides clear statement and strong evidence for most of the claims. I only have a very minor holdback on the claim of "MFSC is the first to incorporate bisimulation into the MVRL framework". The claim is true but the main concern is that the method used to incorporate bisimulation into the MVRL framework are very similar to DBC [1] except the introduction of state aggregator to fuse multi-view observations. I would suggest authors to better discuss the proposed method and DBC in related work or Section 4.1. I think MFSC have many advantages compared to DBC, especially on retaining details with the latent reconstruction task. More discussion on the bisimulation metric part will make the contribution even stronger.


[1] Zhang, A., McAllister, R., Calandra, R., Gal, Y., and Levine, S. Learning invariant representations for reinforcement learning without reconstruction. International Conference on Learning Representation, 2021.

**Essential References Not Discussed:**

No.

**Experimental Designs Or Analyses:**

Yes, the experimental results are sound and valid at my best knowledge.

**Methods And Evaluation Criteria:**

Yes, the proposed methods are well evaluated.

**Other Comments Or Suggestions:**

No.

**Other Strengths And Weaknesses:**

No.

**Questions For Authors:**

Could you share the differences between MFSC and DBC when using the bisimulation metric?

**Relation To Broader Scientific Literature:**

I believe the proposed method pushes further in both the field of multi-view reinforcement learning and the field of robust representation learning for RL.

**Theoretical Claims:**

Yes, they all seem correct.

---

> ### Author Rebuttal · Authors · 2025-03-31
>
> We sincerely thank all the reviewers for their thoughtful evaluation, constructive feedback, and valuable suggestions, which have greatly contributed to improving the clarity, depth, and overall quality of our work. We will address your concerns in the following.
>
> ------
>
>
> **Q: Could you share the differences between MFSC and DBC when using the bisimulation metric?**
>
> We sincerely appreciate the reviewer’s insightful question and are pleased to provide a more detailed clarification of the differences between MFSC and DBC in their use of the bisimulation metric. Specifically, MFSC differs from DBC in two key aspects:
>
> 1. **Avoiding Wasserstein distance for behavioral similarity.** The bisimulation metric generally requires calculating the Wasserstein distance between distributions, which can be computationally expensive. To mitigate this complexity, DBC models latent dynamics transitions as Gaussian distributions, leveraging Euclidean distances to compute a closed-form Wasserstein distance, optimizing an $\ell_1$ distance between representations. However, this approach assumes Gaussian dynamics and introduces an inconsistency between $\ell_1$ and $\ell_2$ (Euclidean) distances, potentially leading to inaccurate approximations. MFSC adopts sampling-based computation approach via independent coupling strategy, encoding current observations into latent representations and explicitly learning latent dynamics to generate representations of subsequent states.
> 2. **Cosine distance as the base metric for latent state representation.**  Inspired by SimSR,  MFSC adopts cosine distance as the base metric for bisimulation computation. Please refer to our response W1 to Reviewer JyV5 for more details regarding the benefit of using cosine distance as the base metric.  Besides, both $\ell_1$ distance (used for reward differences) and cosine distance (used for state differences) are conveniently scalable to matrix operations. Consequently, MFSC can efficiently compute a correlation matrix of state representations within a batch and optimize it to accelerate representation learning. This strategy significantly enhances efficiency compared to DBC’s pairwise comparisons via permutation, resulting in faster and more effective representation updates.
>
> We sincerely thank the reviewer for the valuable comment and will provide a detailed clarification in the appendix of the revised manuscript.
>
> ------
>
> Please do not hesitate to let us know if you have any additional comments.

---

### Official Review · Reviewer_JyV5 · 2025-03-18

**Overall Recommendation:** 3

**Summary:**

This paper addresses key challenges in multi-view reinforcement learning (MVRL), specifically redundancy in observations, distracting or irrelevant information, and robustness to missing views. To overcome these issues, the authors propose a framework, Multi-view Fusion State for Control (MFSC), integrating task-relevant representation learning inspired by the bisimulation metric and a self-attention-based fusion mechanism. Additionally, the method leverages a novel masking-based latent reconstruction auxiliary task to improve robustness against incomplete or missing observations. Experimental validation demonstrates that MFSC consistently achieves superior performance and robustness compared to existing state-of-the-art methods across several robotic manipulation and locomotion benchmarks.

**Claims And Evidence:**

The manuscript provides the following valid claims and their supporting evidences:

**Claim 1:**  MFSC effectively learns compact and task-relevant multi-view representations.

**Evidence 1:** Empirical evaluation across multiple robotic manipulation tasks (Meta-World, PyBullet Ant) shows MFSC significantly outperforms baselines. In addition, visualization (Figure 4) qualitatively analyses using Grad-CAM show the method consistently focuses on task-critical features, indicating successful extraction of task-relevant information.

**Claim 2:** Proposed masking-based latent reconstruction improves robustness against missing views.

**Evidence 2:** Experiments demonstrate MFSC maintains stable and superior performance under scenarios with missing or noisy views, outperforming state-of-the-art methods such as LookCloser and F2C. The authors also provide an ablation test on Figure 7.(b).

**Essential References Not Discussed:**

The manuscript adequately covers relevant literature. To my knowledge, no essential references appear to be missing.

**Experimental Designs Or Analyses:**

The experimental design is comprehensive, and analyses demonstrate method effectiveness and robustness.

**Methods And Evaluation Criteria:**

The proposed methods and evaluation criteria are appropriate and well-justified.

**Other Comments Or Suggestions:**

In the manuscript, the paper title and method name are not intuitively clear about the proposed approach. Emphasizing key components such as bisimulation and self-attention fusion could help better highlight the novelty and differentiating aspects of the research.

**Other Strengths And Weaknesses:**

### Strengths:

**1.	Practical Contribution:**

The paper effectively integrates bisimulation metric learning into multi-view RL settings, particularly utilizing self-attention fusion modules, demonstrating clear practical advantages.

**2.	Sound Experimental Validation:**

The research questions are well-motivated, and the authors provide extensive experimental evidence supporting their approach.


### Weaknesses
**1. Weak Connection between Bisimulation Metrics:**

The theoretical connection between the proposed expected cosine distance and the traditional bisimulation metric (originally employing the Wasserstein distance) is relatively weak and lacks sufficient analysis or justification.

**2. Novelty:**

Both bisimulation metric and self-attention mechanisms themselves are individually well-established techniques, which somewhat limits the standalone novelty of the method. The paper does not clearly emphasize the theoretical advantages or technical challenges arising specifically from integrating these two components. Clarifying how this integration uniquely contributes to MVRL would strengthen the novelty claim.

**Questions For Authors:**

**Question 1.**

Why did the authors consider the additional reconstruction loss necessary? I am slightly concerned that the proposed auxiliary objective (reconstruction loss) may inadvertently encourage the learned representations to rely on redundant information from views that could be missing during inference.

**Question 2.**

Why did the authors choose to compare MFSC against the convergence performance of the specific baseline method (F2C)? Clarification on the motivation behind this comparison method would be helpful.

**Relation To Broader Scientific Literature:**

This work has potential implications for real-world applications requiring control under diverse observational settings, such as autonomous driving. In addition, one particularly relevant application area is multi-modal learning, such as vision-language models (VLM).

**Theoretical Claims:**

The bisimulation metric-motivated representation learning in MFSC ensures that the difference between optimal value functions of the original and latent MDPs is bounded, thus providing a theoretical justification for learning compact yet task-relevant multi-view representations.

However, a theoretical gap exists between the Wasserstein distance between transition distributions—originally used in the MICo update operator—and the expected cosine distance between successor observations employed in this work. Although the authors attribute this gap primarily to computational complexity, a more rigorous theoretical discussion is necessary. Specifically, clarifications on the relationship between two metrics such as how the two metrics differ theoretically, and under which conditions they become equivalent, would significantly strengthen the paper.

---

> ### Author Rebuttal · Authors · 2025-03-31
>
> Thanks for your insightful comments, and we will address your concerns in the following.
>
> ------
>
> **W1:**
>
> We noticed that the citation in Definition 3.1 was incorrect; the correct reference should be the $\pi$-bisimulation metric proposed by Castro et al.(2020)[1], rather than Castro et al.(2021)[2]. We apologize for the confusion caused. To clarify, we provide a brief clarification on the connection between these works:
>
> Conventional bisimulation metric needs to compute the Wasserstein distance over the transition distributions across all actions, which is computationally expensive. Instead, Castro et al.(2020)[1] developed $\pi$-bisimulation which removes the requirement of considering all actions and only needs to consider the actions induced by a policy $\pi$. Castro et al.(2021)[2] introduces independent coupling and proposes a sampling-based metric that does not rely on the Wasserstein distance; however, this comes at the cost of violating the “zero self-distance”,  potentially leading to representational collapse. In contrast, our use of the cosine distance is both efficient and guarantees this key property by design, as also demonstrated in SimSR [3]. This theoretically safeguards against collapse in learned representations. We will elaborate on this point more comprehensively in the next revision.
>
> [1] Castro, et al.(2020). Scalable Methods for Computing State Similarity in Deterministic Markov Decision Processes.
>
> [2] Castro, et al.(2021). MICo: Improved representations via sampling-based state similarity for Markov decision processes.
>
> [3] Zang, H, et al.(2022). SimSR: Simple Distance-Based State Representations for Deep Reinforcement Learning.
>
> ------
>
> **W2:**
>
> We fully agree that both bisimulation metrics and self-attention mechanisms are well-established techniques, yet each has limitations when applied in isolation to MVRL. Specifically, self-attention effectively aggregates information from multiple views but typically lacks an explicit mechanism to focus on task-relevant features, potentially introducing irrelevant or redundant information into the final representations. On the other hand, bisimulation alone does not explicitly utilize the correlations across multiple views. To the best of our knowledge, such an integration has not been explored in the context of MVRL. Following your suggestions, we will carefully revise the manuscript to more clearly emphasize the distinct advantages of this integration.
>
> ------
>
> **The paper title and method name**
>
> Thanks for your point out. We agree that emphasizing core elements such as bisimulation and self-attention would better highlight the novelty of our method. We will consider revising the paper title and method name to more clearly reflect these components by incorporating terms like “Bisimulation-Constrained Attentive”.
>
> ------
>
> **Q1: necessity of the reconstruction loss**
>
> The reconstruction objective is designed to exploit the inherent cross-view dependencies in multi-view observations, thereby enhancing the model’s representation capacity. The effectiveness of this self-supervised objective has been demonstrated in our ablation study (Fig.7b).
>
> We would like to offer some clarifications regarding your concerns. Crucially, the reconstruction objective is applied in the **latent space**, distinct from pixel-level reconstruction. This design encourages the model to reconstruct information that is shared across multiple views, rather than relying on superficial pixel-level redundancy. Moreover, in our downstream RL tasks, we exclusively utilize $z_{L}^{0}$, which is optimized via the bisimulation objective. The theoretical foundation of bisimulation guarantees that $z_{L}^{0}$ preserves task-relevant information aggregated from multi-view observations. For this representation, the reconstruction objective serves to recover task-relevant information. We therefore contend that the representations used for downstream control primarily capture task-relevant features rather than redundant information, as demonstrated by the visualizations in Fig.4\&6. We hope this explanation clarifies our motivations and design choices.
>
> ------
>
> **Q2: results of F2C**
>
> We conducted a careful review of recent MVRL baselines and considered that F2C demonstrates competitive performance on the MetaWorld benchmark according to its performance report. However, the official implementation of F2C for MetaWorld has not been released, and we were unable to reproduce its reported results, despite reaching out to the authors multiple times without receiving any response yet. To enable a comprehensive comparison and study in MVRL, we report the convergence performance of F2C as provided in the original paper. We appreciate your suggestion and will provide a detailed explanation of this decision in the revised version of our paper.
>
> ------
>
> Please do not hesitate to let us know if you have any additional comments.

---

### Decision · Program_Chairs · 2025-05-01

**Decision:**

Accept (poster)

**Comment:**

This paper introduces a novel method for multi-view reinforcement learning (MVRL) that combines bisimulation metric learning with a reconstruction loss. The paper is well-written, includes both theoretical and empirical evaluations, and is in general a solid contribution to the community.

The reviewers unanimously agree on its clarity, quality, and decision to accept.